# The INO80 chromatin remodeller facilitates DNA damage bypass via postreplicative gap repair

Ronald P Wong [1], Mariia Likhodeeva[2], Karl-Peter Hopfner [2] & Helle D Ulrich [1]✉

## Abstract

Chromatin remodellers play a crucial role in transcription, replication and genome maintenance by dynamically regulating the accessibility of DNA. The INO80 complex contributes to multiple aspects of DNA repair and DNA replication stress management. It primarily facilitates nucleosome sliding and has been linked to the removal of the histone variant H2A.Z from chromatin. Our study demonstrates that the INO80 complex promotes DNA damage bypass through postreplicative daughter-strand gap repair. The complex acts downstream of PCNA ubiquitylation, enabling both Exo1-dependent gap expansion and gap filling via translesion synthesis or template switching. Importantly, its function in DNA damage bypass is independent of H2A.Z exchange and separable from its roles in DNA double-strand break repair and transcriptional regulation. Instead, it is consistent with a ruler-like activity that ensures proper nucleosome positioning around both ends of daughter-strand gaps, providing access to exonucleases and DNA polymerases involved in gap processing. These findings underscore the importance of the INO80 complex in facilitating access to damaged DNA regions for timely and effective repair.

**Keywords** DNA Damage Bypass; Chromatin Remodelling; Genome Stability; DNA Replication; Daughter-Strand Gaps
**Subject Categories** DNA Replication, Recombination & Repair; Post-translational Modifications & Proteolysis

## Introduction

Efficient and accurate processing of DNA replication-blocking lesions is essential for genome stability in proliferating cells. Encounter of an unrepaired lesion causes the replisome to stall, which may eventually lead to replication fork breakdown, failure to complete genome duplication, and cell death. To avoid these detrimental effects, cells harbour DNA damage tolerance (DDT) pathways, also known as DNA damage bypass, which facilitate the replication of damaged DNA (Kondratick et al, 2021; Marians, 2018). In eukaryotes, they are largely controlled by the *RAD6* pathway, involving modification of an essential replication factor, PCNA, with the posttranslational modifier ubiquitin (Ripley et al, 2020; García-Rodríguez et al, 2016). Monoubiquitylation of PCNA at a conserved lysine, K164, is mediated by the ubiquitin-conjugating enzyme (E2) Rad6 and the ubiquitin protein ligase (E3) Rad18 (Hoege et al, 2002). The modification activates a mutagenic pathway of translesion synthesis (TLS) via recruitment of damage-tolerant DNA polymerases that can directly synthesise across lesions (Stelter & Ulrich, 2003; Bienko et al, 2005; Plosky et al, 2006). Extension to a K63-linked polyubiquitin chain activates an error-free, recombination-based mechanism known as template switching (TS), which avoids usage of the damaged template altogether by drawing on the information encoded by the undamaged sister chromatid (Hoege et al, 2002; Kondratick et al, 2021).

Early observations in bacteria (Rupp and Howard-Flanders, 1968) and human cells (Lehmann, 1972) suggested that DDT operates at least in part in a postreplicative manner. This was confirmed by numerous studies showing that replication repriming downstream of lesions gives rise to so-called daughter-strand gaps that activate damage signalling (García-Rodríguez et al, 2018) and are filled by TLS or TS, independently of the replisome (Wong et al, 2021, 2020; Tirman et al, 2021; Edmunds et al, 2008). Considering that newly synthesised DNA is rapidly chromatinised after passage of the replisome (Stewart-Morgan et al, 2020), daughter-strand gap filling is likely to occur in a chromatin environment. Yet, the influence of chromatin on DDT is still poorly understood.

Chromatin structure and dynamics are regulated at multiple levels by the identities and posttranslational modifications of histones as well as the positioning, exchange, or removal of nucleosomes by histone chaperones and chromatin remodellers (Eustermann et al, 2024). All these elements have been implicated directly or indirectly in DDT, with varying consequences for TLS and TS (Selvam et al, 2020; Game and Kaufman, 1999; Kats et al, 2009; González-Garrido and Prado, 2023; Dolce et al, 2022; Renaud-Young et al, 2015; House et al, 2014; Wong et al, 2011; Kim et al, 2014; Niimi et al, 2012; Falbo et al, 2009; Kato et al, 2012). Many of the factors facilitating DDT were reported to act at an early stage of the reaction, mainly by enhancing chromatin accessibility, thus facilitating recruitment of Rad18 and promoting PCNA ubiquitylation. Examples include the tumour suppressor protein ING1b (Wong et al, 2011), the transcriptional repressor ZBTB1 (Kim et al, 2014), and chromatin remodellers such as BAF180/Rsc2 (Niimi et al, 2012) and the INO80 complex (Falbo et al, 2009; Kato et al, 2012).

[1]Institute of Molecular Biology, Mainz, Germany. [2]Gene Center, Department of Biochemistry, Ludwig Maximilians University Munich, Munich, Germany. ✉E-mail: h.ulrich@imb-mainz.de

Here, we investigated the mechanistic basis for the contribution of the INO80 complex to DDT. This chromatin remodeller plays diverse roles in transcription, replication and DNA repair (Poli et al, 2017; Morrison, 2017). It mediates nucleosome sliding (Udugama et al, 2011; Brahma et al, 2017) and has been implicated in the exchange of the histone variant H2A.Z from chromatin (Watanabe et al, 2013; Brahma et al, 2017). Importantly, the complex acts in a ruler-like fashion, reading out DNA shape and measuring distances for nucleosome positioning (Basu et al, 2021; Oberbeckmann et al, 2021a, 2021b; Kunert et al, 2022). A role in DNA double-strand break (DSB) repair has been demonstrated, where the INO80 complex supports homologous recombination (HR) (Morrison et al, 2004; van Attikum et al, 2004; Wu et al, 2007), likely by promoting DNA end resection and Rad51 loading (van Attikum et al, 2004; Gospodinov et al, 2011; Nishi et al, 2014; Lademann et al, 2017). The complex also facilitates nucleotide excision repair by enhancing chromatin relaxation prior to incision (Jiang et al, 2010). In addition, the INO80 complex is important for the recovery from DNA replication stress (Shimada et al, 2008; Papamichos-Chronakis and Peterson, 2008) and has been linked to the prevention of transcription-replication conflicts by suppressing pervasive transcription (Xue et al, 2017; Topal et al, 2020) and facilitating the release of RNA polymerase II from chromatin (Lafon et al, 2015; Poli et al, 2016; Prendergast et al, 2020). Its functions in DSB repair and replication stress management have been attributed to its role in H2A.Z removal, as deletion or depletion of the histone variant suppresses HR defects (Alatwi and Downs, 2015; Lademann et al, 2017) and sensitivity to replication stress (Papamichos-Chronakis et al, 2011). A contribution to DDT upstream of PCNA ubiquitylation was postulated in yeast and mammalian cells, mechanistically based on a reported defect in Rad18 recruitment to chromatin (Falbo et al, 2009; Kato et al, 2012).

Upon revisiting these findings, we now discovered that the INO80 complex acts in DDT not upstream, but downstream of PCNA ubiquitylation. By separating the process of lesion bypass from genome replication, we found that the INO80 complex facilitates Exo1-dependent expansion of daughter-strand gaps but also promotes gap filling, resulting in a net contribution to gap repair. Unexpectedly, its function in postreplicative DDT was independent of its potential role in H2A.Z exchange and therefore distinct from its action at DSB ends. In vitro, the INO80 complex was able to reposition nucleosomes away from junction structures mimicking both ends of daughter-strand gaps, providing a possible mechanistic explanation for its effects on gap expansion and filling, respectively. Overall, our study reveals an H2A.Z-independent function of the INO80 complex in the bypass of DNA lesions that appears to be mediated by its ruler-like activity in nucleosome repositioning around postreplicative daughter-strand gaps.

# Results

## The INO80 complex contributes to Rad6-independent damage processing pathways

Considering the reported function of the INO80 complex in the DDT pathway (Falbo et al, 2009; Kato et al, 2012), we examined genetic relationships between INO80 complex mutants and

mutants in ubiquitin-dependent DDT. We observed that deletion of *ARP8*, encoding an INO80 complex subunit, sensitised *rad18Δ* to a range of agents causing DNA damage and replication stress, such as the alkylating compound methyl methanesulfonate (MMS), the fork-stalling agent hydroxyurea (HU), and ultraviolet (UV) irradiation (Fig. 1A). This confirmed that the INO80 complex has additional functions in genome maintenance independent of the *RAD6* pathway. Additive effects on damage sensitivity were also observed upon degron-mediated depletion of two other subunits of the complex, Ino80 and Arp5, in the absence of Rad18 (Fig. EV1A,B).

To differentiate between effects on TS *versus* TLS, we deleted *ARP8* in a *ubc13Δ* background, selectively defective in TS (Hoege et al, 2002), and in a mutant devoid of any TLS polymerase activity, *tlsΔ* (*rad30Δ rev1Δ rev3Δ*). We found a strong enhancement of damage sensitivity in both *ubc13Δ arp8Δ* and *tlsΔ arp8Δ* in response to MMS and UV, suggesting that the loss of functional INO80 complex impacts on both branches of the *RAD6* pathway (Fig. 1A). Notably, such synergy was not observable in the presence of HU, which differs from MMS and UV in that it activates damage signalling at the stalled fork, while the damage signal in response to MMS or UV emerges largely from postreplicative daughter-strand gaps (García-Rodríguez et al, 2018). Specific consequences for TS were monitored in *pol32Δ* and *smc6-56* mutants. TS-specific defects, such as in *ubc13Δ*, have been reported to suppress the cold sensitivity of *pol32Δ* and the damage sensitivity of *smc6-56* mutants (Karras and Jentsch, 2010; Choi et al, 2015). Unlike *ubc13Δ*, however, deletion of *ARP8* enhanced both phenotypes (Fig. 1B,C). Similar effects were observed upon depletion of the catalytic subunit, Ino80 (Fig. EV1C,D). Notably, the double mutants *rad18Δ arp8Δ, pol32Δ arp8Δ*, and *smc6-56 arp8Δ* exhibited strong synthetic growth defects compared to the respective single mutants (Fig. 1A–C), consistent with a previous screen for synthetic fitness defects (Pan et al, 2006).

These genetic data suggest that the overall genome stability phenotype of INO80 complex mutants is pleiotropic and is dominated by Rad6-independent aspects. Yet, they do not exclude a contribution to ubiquitin-dependent DDT and may indicate a particular sensitivity to situations where daughter-strand gap repair is partially impeded.

## The INO80 complex acts downstream of PCNA ubiquitylation in DDT

A previous study had reported a PCNA ubiquitylation defect in INO80 mutants, arising from inefficient Rad18 recruitment (Falbo et al, 2009). Those experiments had involved either continuous exposure to 0.02% MMS throughout S phase or a combination of MMS and HU, and consequently very high damage loads. We therefore considered the possibility that the general proliferation defect of INO80 complex mutants and their Rad6-independent damage sensitivity might have caused fork collapse or prevented efficient replication altogether. In that case, conditions would not have been ideal to observe Rad18 recruitment or PCNA ubiquitylation. We therefore revisited the PCNA ubiquitylation response under conditions of tolerable damage loads, induced by a single 30 min pulse of the same MMS concentration used in the previous study (0.02%) in G1-synchronised populations. Under these conditions, PCNA ubiquitylation upon release into S phase is

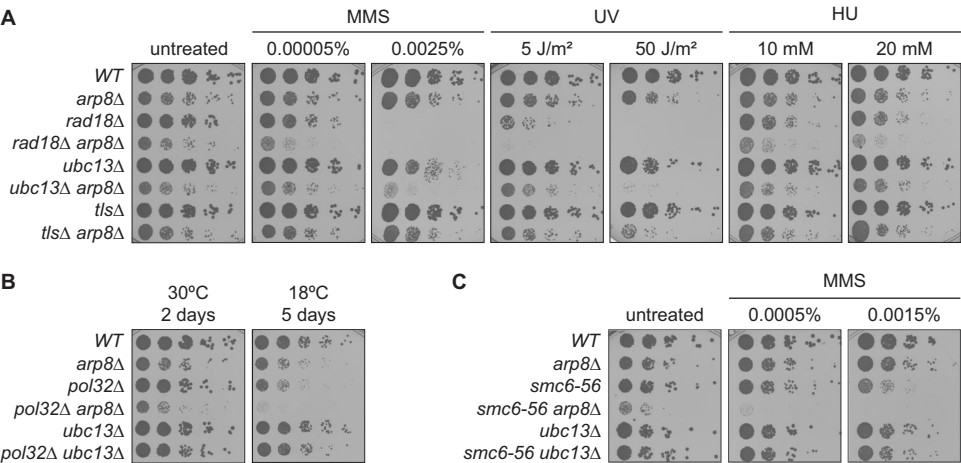

**Figure 1. The INO80 complex contributes to *RAD6*-independent pathways of damage resistance.**

(A) Deletion of *ARP8* sensitises *RAD6* pathway mutants *rad18Δ*, *ubc13Δ* and *tlsΔ* (*rev1Δ rad30Δ rev3Δ*) towards DNA-damaging agents. Colony survival was assessed on plates containing the indicated agents or UV-irradiated before incubation (biological replicates: $N = 1$). (B, C) *ARP8* does not act specifically in the TS pathway. Suppression of the cold sensitivity of *pol32Δ* by *arp8Δ* was assessed by growth at the indicated temperatures (B; biological replicates: $N = 2$), and suppression of the damage sensitivity of *smc6-56* by *arp8Δ* was assessed by growth in the presence of MMS (C; biological replicates: $N = 2$). A *ubc13Δ* mutant served as control for a TS-specific defect. Source data are available online for this figure.

mostly associated with the postreplicative filling of daughter-strand gaps resulting from repriming of the replisome downstream of the lesions (Wong et al, 2020). In *ino80Δ* and *arp8Δ* mutants, we found that PCNA ubiquitylation was not reduced but merely delayed, in line with a slower progression through S phase (Fig. EV2A,B).

To avoid any interference from the strong general growth defect of INO80 complex mutants, we then used a degron-tagged allele of *INO80* (*INO80^{AID*-FLAG}*) for transient depletion of the catalytic subunit during a single cell cycle (Fig. 2A). In this setting, the delay in cell cycle progression in the absence of Ino80 with or without DNA damage was minimal (Figs. 2B and EV2C,D). Yet, passage through S phase after MMS treatment resulted in prolonged checkpoint activation, monitored by the appearance of a smear of high-molecular weight species of Rad53, representing its various phosphorylated forms (Fig. 2B), and prolonged PCNA ubiquitylation (Fig. 2C). Moreover, resolution of DNA damage-induced Replication Protein A (RPA) foci was strongly compromised (Figs. 2D and EV2E). We had previously shown that RPA foci represent DNA damage-induced daughter-strand gaps that arise from replisome repriming downstream of lesions in the replication template and are resolved by the *RAD6* pathway (Wong et al, 2020). Therefore, these results indicate that—at least in response to tolerable levels of DNA damage—the INO80 complex contributes to efficient DDT not upstream, but downstream of PCNA ubiquitylation. It is possible that the reported defects in Rad18 recruitment and, consequently, PCNA ubiquitylation (Falbo et al, 2009) may have resulted from the general replication problems of the deletion mutants.

## The INO80 complex facilitates both DNA resection and daughter-strand gap repair

Direct evidence for an accumulation of daughter-strand gaps in the absence of the INO80 complex was obtained by quantifying ssDNA

within EdU-labelled regions of newly replicated DNA using fibre assays (Fig. 3A,B). We found that even without exogenous damage, depletion of Ino80 caused a ca. threefold increase in both the fraction of ssDNA (from 5 to 14%) and the tract density, i.e., the number of ssDNA tracts per μm (from 0.06 to 0.21 μm$^{-1}$) (Figs. 3C,D and EV3A–C). This indicated an overall higher level of spontaneous replication problems in the absence of Ino80. As expected, treatment with MMS enhanced the total amount of ssDNA and the density of tracts compared to the undamaged samples. Interestingly, however, depletion of Ino80 under damage conditions resulted in an increased fraction of ssDNA, but no significant difference in the number of tracts per μm (Fig. 3C,D and EV3B,C). This suggested that ssDNA tracts are longer in the absence of Ino80.

Longer regions of ssDNA might arise from either enhanced expansion or a defect in the filling of daughter-strand gaps. To distinguish between these scenarios, we first examined whether the INO80 complex counteracts gap expansion (Fig. 3E). We previously showed that expansion of daughter-strand gaps is due to Exo1-mediated resection and contributes to efficient checkpoint activation in early S phase (García-Rodríguez et al, 2018). However, Ino80 depletion resulted in a delay rather than an acceleration of checkpoint activation in response to MMS (Fig. 3F) as well as other DNA-damaging agents (Fig. EV3D). We also observed lower numbers and intensities of RPA foci in early S phase, indicating a reduced amount of total ssDNA (Figs. 3G and EV3E). This suggested that the INO80 complex promotes rather than opposes gap expansion. However, Ino80 depletion also consistently slowed down entry into S phase (Figs. 3F,G and EV3D), making it difficult to decisively conclude an activating role in gap expansion. We therefore monitored the effects of the INO80 complex on Exo1-mediated resection in a different context. In a temperature-sensitive mutant of the telomere-binding protein, Cdc13 (*cdc13-1*), a shift to the non-

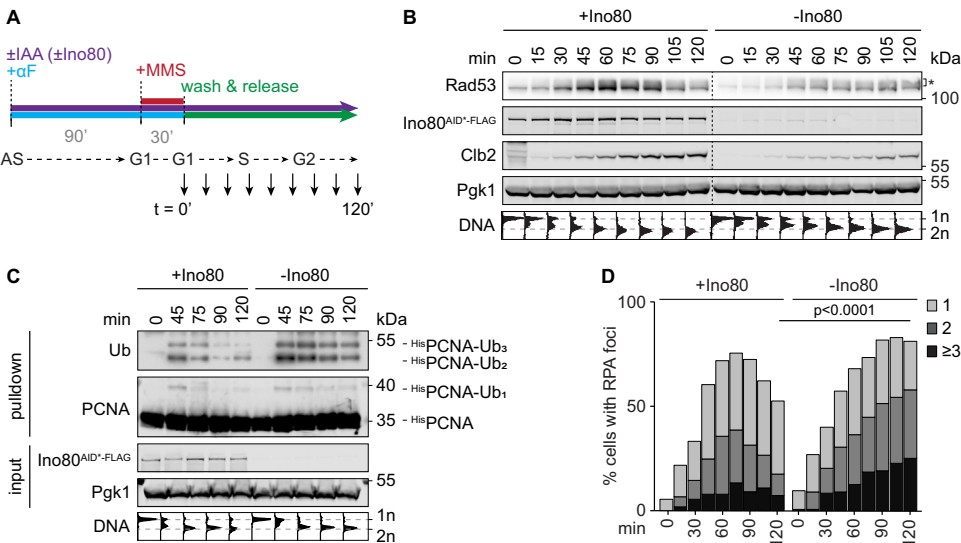

**Figure 2. The INO80 complex prevents the accumulation of ssDNA during replication of damaged DNA.**

(A) Experimental scheme. Cells harbouring *INO80^AID*-FLAG* are synchronised in G1 in the presence or absence of auxin (IAA, 0.5 mM), damaged with 0.02% MMS for 30 min, washed and released into YPD medium without MMS. Samples are collected throughout the subsequent cell cycle for preparation of total lysates, flow cytometry, fluorescence microscopy, and/or Ni-NTA chromatography. (B) Depletion of Ino80 prolongs checkpoint activation as monitored by Rad53 phosphorylation in total lysates, detectable as a smear (*) above the band corresponding to unmodified Rad53. Clb2 served as a cell cycle marker and Pgk1 as loading control. Corresponding cell cycle profiles are shown below the blots (biological replicates: $N = 2$). (C) Depletion of Ino80 causes prolonged PCNA ubiquitylation. Modified PCNA was detected after Ni-NTA pull-down of $^{His}$PCNA from total lysates prepared under denaturing conditions. Corresponding cell cycle profiles are shown below the blots (biological replicates: $N = 1$). (D) Depletion of Ino80 interferes with the resolution of RPA foci. Samples were subjected to fluorescence microscopy to detect Rfa1^GFP. Quantification was performed after segmentation of foci as shown in Fig. EV2E (Mann–Whitney $U$-test) (biological replicates: $N = 1$). Source data are available online for this figure.

permissive temperature results in deprotection and Exo1-dependent resection of the telomere ends, exposure of extended regions of ssDNA, and consequently checkpoint activation (Dewar and Lydall, 2010). In this system, we found that depletion of Ino80 also interfered with efficient checkpoint activation (Fig. 3H). In combination, our data suggest that the INO80 complex generally promotes rather than prevents resection, consistent with its reported action during HR-mediated DSB repair (van Attikum et al, 2004; Gospodinov et al, 2011; Lademann et al, 2017).

We then asked whether the net accumulation of ssDNA in the absence of Ino80 was due to a defect in gap repair (Fig. 3I). We made use of a system where this reaction can be separated from genome replication by controlling *RAD18* expression via a tetracycline-repressible promoter (Daigaku et al, 2010). Here, exposure of G1-synchronised cells to damage and subsequent passage through S phase in the absence of Rad18 causes an accumulation of daughter-strand gaps and a checkpoint-induced arrest in G2/M, which is resolved by postreplicative gap filling upon re-expression of *RAD18*. The efficiency of gap filling can be monitored by exit from the G2/M arrest, deactivation of the checkpoint, resolution of RPA foci, and incorporation of a labelled nucleotide, BrdU, as a direct measure of DDT-associated DNA synthesis. When re-expression of *RAD18* in this set-up was combined with depletion of Ino80 upon G2/M arrest (Fig. 3J), gap filling was measurably delayed by all four criteria (Fig. 3K–N). A negative effect of auxin treatment, used for Ino80 depletion, on the efficiency of gap filling was ruled out (Fig. EV3F). Importantly, the onset of PCNA ubiquitylation upon RAD18 re-expression was indistinguishable in the presence or absence of Ino80, but the modification persisted for a longer period (Fig. 3O).

Taken together, our data are consistent with a dual function of the INO80 complex at daughter-strand gaps: on the one hand, the remodeller appears to facilitate gap expansion by promoting resection. On the other hand, it functions downstream of PCNA ubiquitylation in gap filling. Overall, the gap-filling defect must outweigh the decreased gap expansion rates, as the two opposing effects result in a net accumulation of ssDNA in the absence of the INO80 complex.

## The gap-filling function of the INO80 complex is unrelated to H2A.Z removal and transcription

We next investigated the mechanism by which the INO80 complex facilitates gap repair. First, we depleted individual subunits of the complex representing defined structural and functional modules (Tosi et al, 2013) (Fig. 4A,B). While the Arp8 and Arp5 modules cooperate with Ino80 in DNA length sensing and regulating nucleosome positioning (Brahma et al, 2018), the yeast-specific Nhp10 module, composed of Nhp10, Ies1, Ies3 and Ies5, is not critical for nucleosome movement or ATP hydrolysis (Shen et al, 2003) but exhibits strong DNA- and nucleosome-binding properties on its own (Shen et al, 2003; Zhou et al, 2018) and promotes recruitment of the INO80 complex to phosphorylated H2A (γH2A) in the context of DSB repair (Morrison et al, 2004). We found that in contrast to Arp8 and Arp5, Nhp10 was dispensable for efficient resolution of RPA foci (Fig. 4C).

We then asked whether the INO80 complex acts in DDT by removing the histone variant, H2A.Z, from chromatin (Fig. 4D). This activity promotes HR at DSBs and in response to replication stress, and the respective defects of *ino80* mutants can be rescued by deletion of *HTZ1*, the gene encoding H2A.Z (Papamichos-

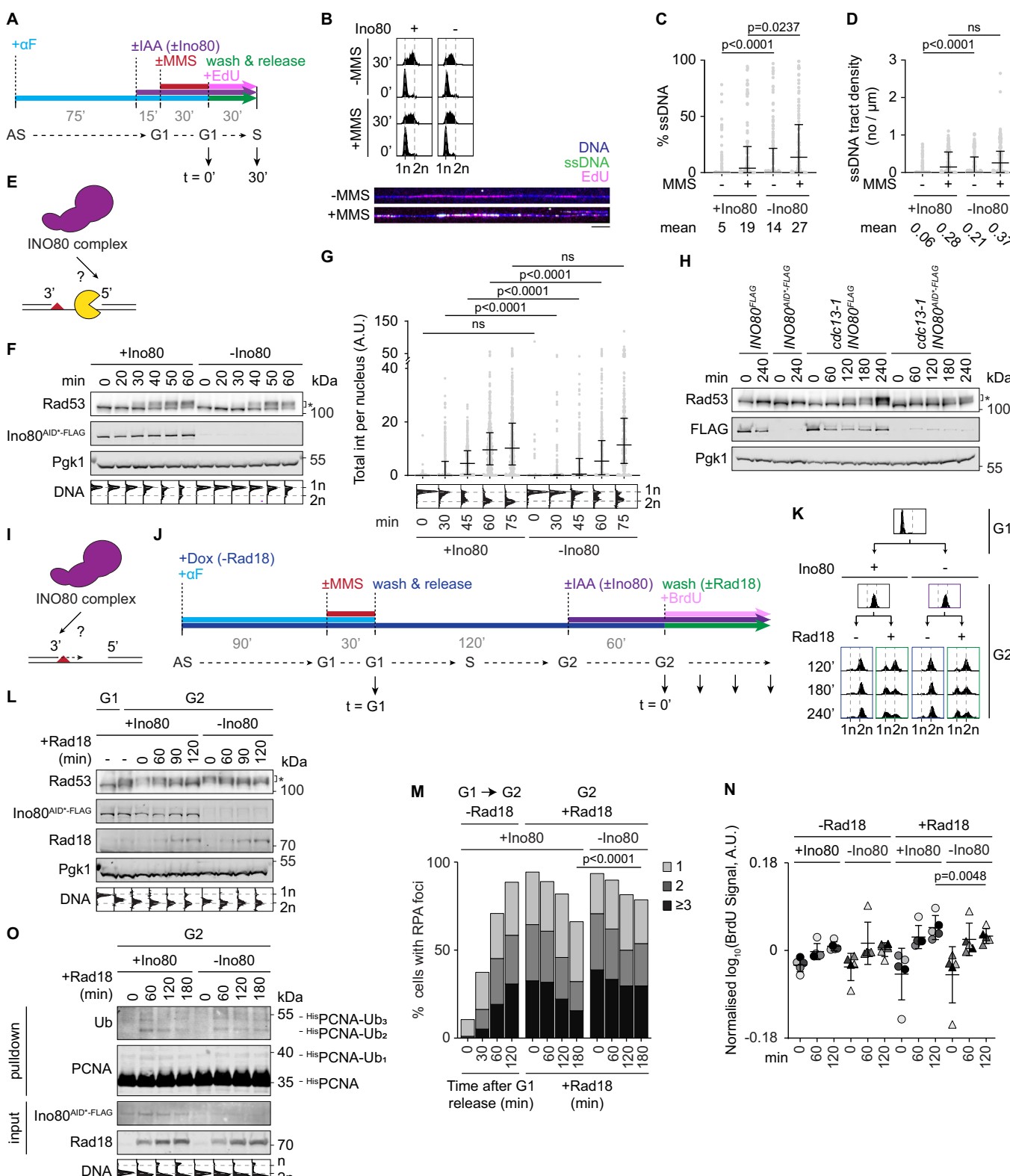

**Figure 3. The INO80 complex facilitates both DNA resection and daughter-strand gap filling.**

(A) Experimental scheme. Cultures are synchronised in G1, treated for 45 min with 1 mM auxin (IAA) for Ino80 depletion where indicated, damaged with 0.02% MMS for 30 min where indicated, washed and released into S phase in the presence of EdU. Samples are collected 30 min after release for flow cytometry, genomic DNA extraction, and DNA combing. (B) DNA damage-induced ssDNA accumulates in newly replicated DNA. Top: Cell cycle profiles of cells treated according to panel A (biological replicates: $N = 1$). A biological replicate is shown in Fig. EV3A. Bottom: representative images of native DNA fibres collected from *BrdUx7 INO80^{AID*-FLAG}* cells in the presence of Ino80. Scale bar: 10 μm. (C) Depletion of Ino80 enhances the fraction of ssDNA within newly replicated (EdU-labelled) DNA (Mann–Whitney *U*-test; biological replicates: $N = 2$). Bars indicate median with interquartile ranges. A biological replicate is shown in Fig. EV3B. (D) Depletion of Ino80 does not enhance the number of ssDNA tracts per length of newly replicated (EdU-labelled) DNA after MMS treatment (Mann–Whitney *U*-test; ns not significant; biological replicates: $N = 2$). Bars indicate median with interquartile ranges. A biological replicate is shown in Fig. EV3C. (E) Schematics of a potential role of the INO80 complex in daughter-strand gap expansion. The lesion is represented as a red triangle, and Exo1 is shown in yellow. (F) Depletion of Ino80 delays damage-induced checkpoint activation (*) but also S phase progression. *TET-RAD18 INO80^{AID*-FLAG}* cells were treated as in Fig. 2A for cell cycle and checkpoint analysis (biological replicates: $N = 2$). (G) Depletion of Ino80 delays accumulation of damage-induced ssDNA. The total intensity of Rfa1^{GFP} per nucleus was quantified after treatment as in Fig. 2A. The corresponding percentage of cells with Rfa1^{GFP} foci is shown in Fig. EV3E (Mann–Whitney *U*-test; ns not significant; biological replicates: $N = 2$). Bars indicate median with interquartile ranges. (H) Depletion of Ino80 delays checkpoint activation upon telomere uncapping. The indicated strains were grown for 2 h at 23 °C in the presence of nocodazole to induce G2/M arrest, followed by the addition of auxin for 30 min to deplete Ino80^{AID*-FLAG}. Cells were then shifted to 37 °C ($t = 0$) and samples were collected at the indicated time points for western blot analysis (*: Rad53 phosphorylation; biological replicates: $N = 1$). (I) Schematics of a potential role of the INO80 complex in daughter-strand gap filling. (J) Experimental scheme. *TET-RAD18 INO80^{AID*-FLAG}* cells grown in the presence of doxycycline to repress *RAD18* expression are synchronised in G1, damaged with 0.033% MMS for 30 min ($t = G1$), and released into S phase. Two hours after release, when cells have accumulated in G2/M phase, 1 mM auxin (IAA) is added to degrade Ino80 for 1h before doxycycline is washed off to re-express *RAD18* ($t = 0$). To quantify newly synthesised DNA, BrdU was added at the time of doxycycline wash-out. (K) Activation of DDT in G2/M-arrested cells allows re-entry into the cell cycle in the presence or absence of Ino80, but re-entry is delayed in the absence of Ino80. Nocodazole was omitted for this experiment to allow progression to the next G1 phase (biological replicates: $N = 3$). (L) Depletion of Ino80 delays checkpoint deactivation during postreplicative gap filling. Note that abundant phosphorylation of Rad53 (*) appears at the expense of the unphosphorylated form (biological replicates: $N = 2$). A biological replicate is shown in Fig. EV3G. (M) Depletion of Ino80 delays RPA foci resolution during postreplicative gap filling (Mann–Whitney *U*-test; biological replicates: $N = 2$). A biological replicate is shown in Fig. EV3H. (N) Depletion of Ino80 delays gap filling as measured by BrdU incorporation into newly synthesised DNA. BrdU was added at the time of *RAD18* re-expression in G2/M ($t = 0$). BrdU intensities (log₁₀-transformed) were corrected by background subtraction and normalised to the mean values of each biological replicate (paired *t*-test; $N = 5$). Bars indicate the mean with standard deviations. Data points are grey-shaded according to the replicates. (O) Depletion of Ino80 prolongs PCNA ubiquitylation during postreplicative gap filling. PCNA modification was detected by Ni-NTA pulldown from lysates of a ^{His}POL30 strain prepared under denaturing conditions (biological replicates: $N = 1$). Source data are available online for this figure.

Chronakis et al, 2011; Brahma et al, 2017; Lademann et al, 2017; Alatwi and Downs, 2015). However, neither the *htz1Δ* mutant nor a deletion of *SWR1*, encoding a remodeller that is responsible for H2A.Z deposition, suppressed the hyperaccumulation of ssDNA during S phase (Fig. 4E and EV4A) or the delay in postreplicative gap filling in G2/M upon depletion of Ino80 (Fig. 4F and EV4B). Interestingly, a slight rescue of gap filling efficiency in G2/M was observed in some instances in the *htz1Δ swr1Δ* double mutant (Fig. EV4B,C). As an independent approach, we also monitored the effect of the *htz1Δ swr1Δ* double deletion on Rad53 phosphorylation but detected no suppression of the prolonged checkpoint activation upon depletion of Ino80 (Fig. EV4D). Together, these results argue against an involvement of H2A.Z exchange in the contribution of Ino80 to gap filling.

Another function of the INO80 complex is the silencing of pervasive transcription. Here, the complex cooperates with the transcription regulators, Mot1 and NC2 (Xue et al, 2017), and combined depletion of Ino80 and Mot1 was reported to cause DSBs at replication origins under conditions of replication stress (Topal et al, 2020). We therefore asked if these conditions would also interfere with gap filling, monitored by elevated levels of ssDNA during replication of damaged DNA. However, due to the strong delay in S phase progression and the delayed appearance of RPA foci observed under these conditions (Topal et al, 2020) (Fig. EV4E,F), these experiments remained inconclusive.

Finally, the INO80 complex has been implicated in RNA polymerase II removal and degradation to prevent transcription-replication conflicts (Lafon et al, 2015; Prendergast et al, 2020). We therefore tested if transient depletion of RNA polymerase II would rescue the replication problems associated with loss of Ino80. Interestingly, depletion of the large subunit of the polymerase, Rpb1, alone resulted in a reduced number of RPA foci compared to an

unperturbed situation (Fig. EV4G,H). Co-depletion of Ino80 and Rpb1 led to a severe S-phase progression defect, which complicated the interpretation of the data (Fig. EV4G,H). However, it clearly did not rescue the replication defect conferred by depletion of Ino80.

In summary, the data presented in this section do not provide evidence for an involvement of H2A.Z exchange, silencing of pervasive transcription, or removal of RNA polymerase II in promoting postreplicative gap filling via the INO80 complex.

## The INO80 complex associates with ssDNA in cells

To determine whether the INO80 complex is physically present at daughter-strand gaps, we used proximity ligation assays (PLAs) to probe the association of FLAG-tagged Ino80 with myc-tagged Rfa1 as a marker of ssDNA. A PLA signal was indeed detectable (Fig. 5A). Its intensity grew as cells progressed into S phase and was prolonged in cells where Rad18 had been depleted and gaps accumulated (Figs. 5B and EV5A). This pattern is consistent with an association of the INO80 complex with daughter-strand gaps marked by RPA during and after S phase. A direct involvement of the INO80 complex in gap processing is therefore likely.

## The INO80 complex repositions nucleosomes away from ds-ssDNA junctions

Previously, the INO80 complex had been shown to move nucleosomes away from features on the DNA that act as barriers, such as DSBs or DNA-bound proteins, and place them at certain distances from such structures (Krietenstein et al, 2016; Oberbeckmann et al, 2021a, 2021b; Kunert et al, 2022). To test whether this repositioning activity was relevant for gap filling, we examined the action of purified INO80 complex on nucleosome substrates

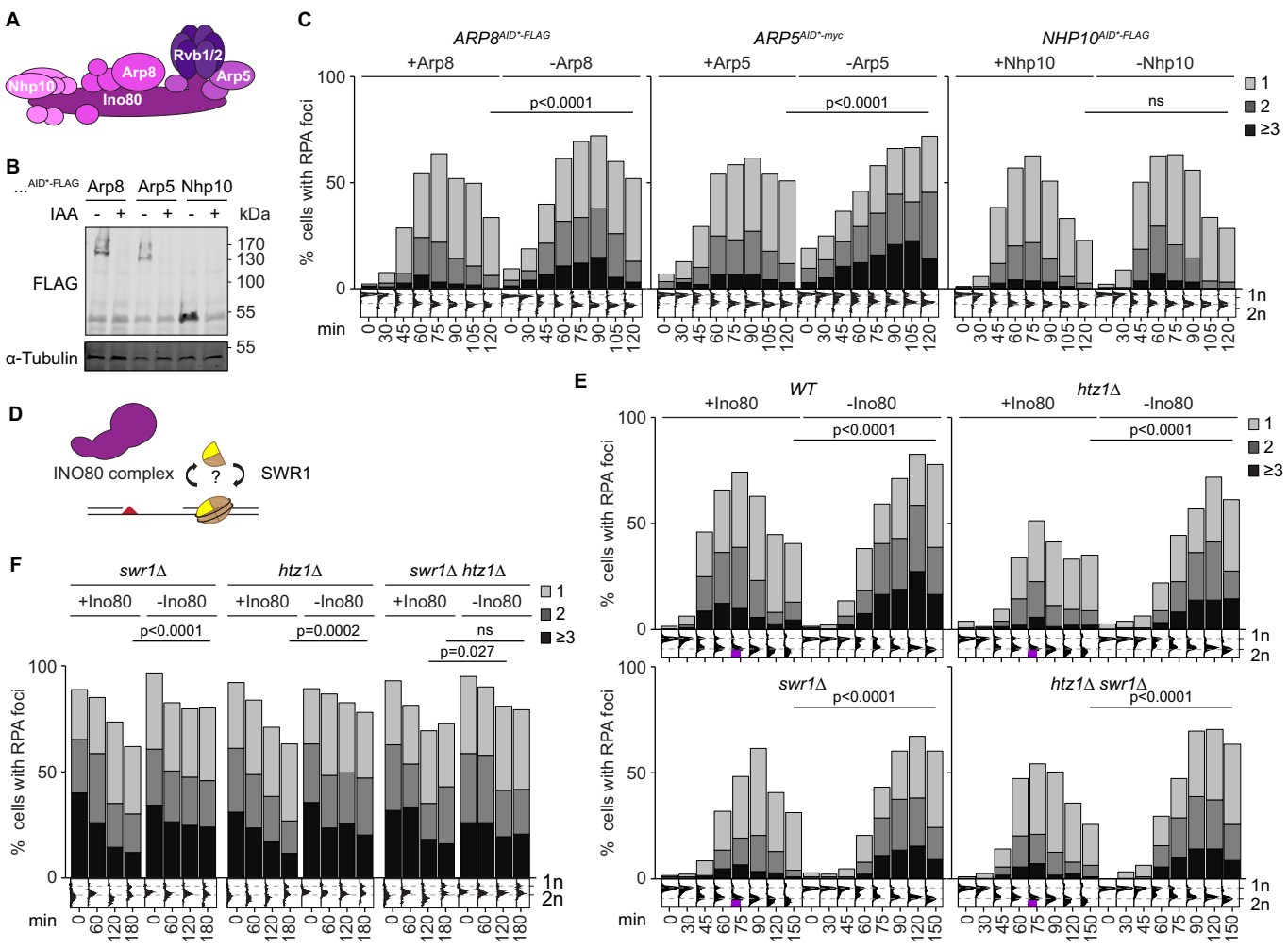

**Figure 4. The function of the INO80 complex in DDT is unrelated to its role in H2A.Z extraction.**

(A) Schematics of the INO80 complex architecture, colour-coded by the individual modules. (B) INO80 complex subunits can successfully be depleted using auxin-inducible AID*-FLAG degrons via treatment with 0.5 mM auxin (IAA) for 120 min (biological replicates: $N = 1$). (C) The Arp8 and Arp5 modules, but not the Nhp10 module, are required for efficient resolution of damage-induced RPA foci. Cells harbouring *RFA1^GFP* and AID*-FLAG-tagged alleles of relevant INO80 complex subunits were treated as described in Fig. 2A in the presence and absence of auxin, and RPA foci were quantified by fluorescence microscopy (Mann–Whitney *U*-test; ns not significant; biological replicates: $N = 1$). Cell cycle profiles are shown below the graphs. (D) Schematics of a potential role of the INO80 complex in histone exchange at ssDNA gaps. Histone H2A.Z is shown in yellow. (E) Deletion of *HTZ1* or *SWR1* does not suppress the defect in replication-associated RPA foci resolution in the absence of Ino80. RPA foci during a cell cycle in the presence or absence of Ino80 were monitored in the indicated strain backgrounds according to the scheme in Fig. 2A (Mann–Whitney *U*-test; biological replicates: $N = 2$). Cell cycle profiles are shown below the graphs. A biological replicate is shown in Fig. EV4A. (F) Deletion of *HTZ1* or *SWR1* does not suppress the defect in postreplicative daughter-strand gap filling in the absence of Ino80. RPA foci resolution in G2/M was monitored in the indicated strain background according to the scheme in Fig. 3J (Mann–Whitney *U*-test; ns not significant; biological replicates: $N = 3$). Cell cycle profiles are shown below the graphs. Two biological replicates are shown in Fig. EV4B,C. Source data are available online for this figure.

mimicking the boundaries of a daughter-strand gap. We argued that the junctions between double- and single-stranded DNA (ds-ssDNA) that delineate a daughter-strand gap might act as such barriers and might prompt the INO80 complex to reposition proximal nucleosomes (Fig. 5C). To mimic the ds-ssDNA junctions, we generated double-stranded DNA substrates containing either a 5′- or a 3′-overhang of 38 nucleotides, representing the "left" and "right" boundary of a daughter-strand gap, respectively (Fig. EV5B–E). An analogous blunt-ended dsDNA substrate served as a control. A histone octamer was assembled close to the junctions or one end of the dsDNA using a strong nucleosome positioning sequence, Widom 601 (Polach and Widom, 1995). All

substrates were incubated in the presence of ATP with recombinant *S. cerevisiae* INO80 complex purified from insect cells. As previously shown (Kunert et al, 2022), a nucleosome next to a blunt dsDNA end was moved from the end to a central position (Fig. 5D). Notably, we observed similar activity on nucleosomes assembled close to a 5′- or 3′-overhang (Fig. 5D). Quantification showed that the repositioning was slower and less efficient, particularly for the 5′-overhang (Fig. 5E). Consistent with our in vivo data, the Nhp10 module was dispensable for nucleosome repositioning, both at blunt ends and at single-stranded overhangs (Fig. 5D,E). In fact, as observed previously (Zhou et al, 2018), recombinant INO80 complex lacking this module consistently

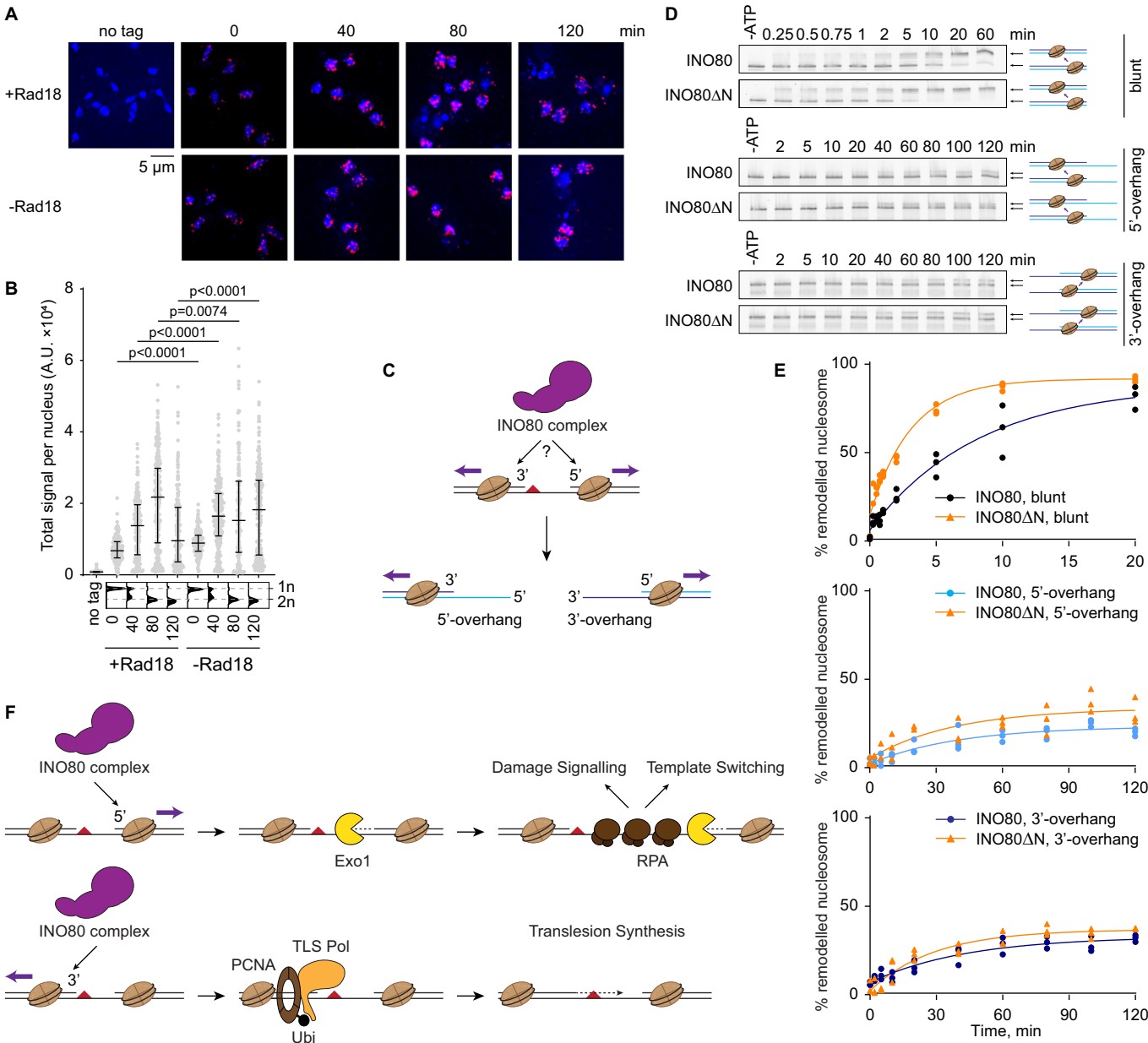

**Figure 5. The INO80 complex associates with ssDNA in cells and slides nucleosomes away from both ends of ds-ssDNA junctions in vitro.**

(A, B) The INO80 complex is present at RPA-covered daughter-strand gaps. PLA signals (red) are detectable with anti-FLAG and anti-myc antibodies in the nuclei (DAPI, blue) of cells harbouring Ino80$^{FLAG}$ and Rfa1$^{9myc}$ upon synchronisation in G1, treatment with 0.02% MMS for 30 min, and release into S phase for the indicated times (A). Cells harbouring untagged alleles served as negative controls. Scale bar: 5 μm. PLA signals were quantified by total nuclear intensity (B). Cell cycle profiles are shown below the graph (Mann–Whitney U-test; biological replicates: $N = 3$). Bars indicate median with interquartile ranges. Two biological replicates are shown in Fig. EV5A. (C) Schematics of a potential role of the INO80 complex in repositioning nucleosomes away from the boundaries of daughter-strand gaps and design of remodelling substrates for in vitro sliding assays. (D, E) Recombinant INO80 complex repositions nucleosomes away from ds-ssDNA junctions independently of the Nhp10 module. ATP-dependent sliding was monitored in vitro by native polyacrylamide gel electrophoresis, using recombinant INO80 or INO80ΔN complex and nucleosomes assembled on dsDNA either with blunt ends or containing a 38 nt single-stranded 5'- or 3'-overhang as indicated (D). The percentage of repositioning at each time point was quantified from three independent experiments (E; biological replicates: $N = 3$). Lines represent one-phase decay curves fitted to the data. Design and preparation of the DNA substrates are shown in Fig. EV5B–E. (F) Model of a contribution of the INO80 complex to DDT. By repositioning nucleosomes away from the boundaries of daughter-strand gaps, the INO80 complex can enhance access to the 5'-terminus of the junction, thus facilitating gap repair by initial gap expansion via Exo1-mediated resection, which in turn promotes checkpoint activation and subsequent gap filling via TS. Alternatively, enhanced access to the 3'-terminus of the junction facilitates TLS. Source data are available online for this figure.

exhibited somewhat higher activities than the full complex. These results demonstrate that both junctions are recognised as barriers for nucleosome positioning. We therefore propose that the INO80 complex may act in postreplicative DNA damage bypass by promoting access to the gap boundaries.

# Discussion

## The INO80 complex acts as a facilitator of daughter-strand gap repair

Our observations suggest a model where the INO80 complex promotes efficient DNA damage bypass by conferring accessibility to both boundaries of a daughter-strand gap (Fig. 5F). On the "right" side of the gap, displaying a free 5′-end, a repositioning of nucleosomes away from the end would initially facilitate resection by a 5′-3′ nuclease such as Exo1 (Vanoli et al, 2010; García-Rodríguez et al, 2018). The resulting gap expansion would, in turn, allow checkpoint activation and would promote subsequent gap repair via the error-free TS pathway. On the "left" side of the gap, a shifting of nucleosomes away from the free 3′-end could provide access for a DNA polymerase to carry out repair synthesis and promote gap filling without prior gap expansion. In this manner, the INO80 complex could stimulate TLS but also error-free gap filling during TS. This dual activity would reconcile our seemingly contradicting observations that the INO80 complex promotes resection, but at the same time, its mutants exhibit delays in daughter-strand gap repair. It would also provide an explanation for the synergistic effects of ino80 mutants with both TLS and TS specifically in situations where damage bypass predominantly occurs in daughter-strand gaps, but not in response to fork-associated replication stress (Fig. 1A).

Our model is compatible with various prior observations. The biochemical activity of the INO80 complex at ds-ssDNA junctions resembles its ruler-like mode of operation by which nucleosomes are moved away from other barrier-like features (Oberbeckmann et al, 2021a, 2021b; Kunert et al, 2022) and spacing of nucleosome arrays is accomplished (Udugama et al, 2011; Zhou et al, 2018; Brahma et al, 2018). On a molecular level, it is likely that ssDNA is not recognised as proper extranucleosomal DNA by INO80's A module (Kunert et al, 2022). Furthermore, nucleosomes are known to impede Exo1 activity in vitro (Adkins et al, 2013). Facilitating Exo1-mediated resection of 5′-ends thus appears to be a common property of the INO80 complex, applying not only in the context of HR-mediated DSB repair (van Attikum et al, 2004; Gospodinov et al, 2011; Lademann et al, 2017) but, as we now show, also at daughter-strand gaps and unprotected telomeres. On the opposite end of the gap, TLS or other modes of repair synthesis would require a DNA polymerase to associate with the ds-ssDNA junction. Considering that most DNA polymerases, including TLS enzymes (Freudenthal et al, 2010; Lancey et al, 2021), act in complex with PCNA, they would be expected to require a sufficiently large stretch of nucleosome-free DNA to engage with the recessed 3′-OH end. Notably, the footprint of human TLS polymerase κ in complex with ubiquitylated PCNA amounts to ca. 20 bp upstream of the 3′-OH end (Lancey et al, 2021), which is well within the range of space generated by the INO80 complex (Udugama et al, 2011). Finally, the reported preference of isolated

Arp8 for ssDNA over dsDNA (Osakabe et al, 2014) could potentially indicate a recruitment mechanism of the INO80 complex to daughter-strand gaps or ssDNA exposed at stalled replication forks.

## The function of the INO80 complex in DDT is distinct from its role in DSB repair

Stimulation of Exo1-dependent resection by the INO80 complex appears to be common to both postreplicative DDT and HR-mediated DSB repair. Yet, we observed major differences between the two settings that indicate distinct mechanisms. Unlike DSB repair (van Attikum et al, 2004; Lademann et al, 2017), processing of replication-stalling lesions by DDT does not require the Nhp10 module of the INO80 complex (Fig. 4C). A possible explanation could be the way in which the remodeller is recruited to its respective target sites. At DSBs, checkpoint activation results in phosphorylation of H2A, which is directly recognised by the Nhp10 module and may therefore serve as a recruitment signal (Morrison et al, 2004; van Attikum et al, 2004; Downs et al, 2004). In contrast, daughter-strand gaps are characterised by an accumulation of ssDNA and RPA phosphorylation (García-Rodríguez et al, 2018; Wong et al, 2021). Whether ssDNA-binding by the Arp8 module is sufficient for targeting the INO80 to daughter-strand gaps or whether additional, potentially phosphorylation-dependent signals are required, is currently unknown. Another difference between DSB repair and DDT is the relevance of H2A.Z exchange. While this activity is important for HR-mediated DSB repair (Alatwi and Downs, 2015; Lademann et al, 2017), we did not find any evidence for an impact of H2A.Z itself or its deposition factor, Swr1, on daughter-strand gap filling. Interestingly, hyper-incorporation of H2A.Z in an ino80Δ mutant was reported to also result in spontaneous DNA replication problems and a sensitisation towards fork stalling (Papamichos-Chronakis et al, 2011). However, given that deletion of HTZ1 did not rescue the DDT defect observed in the absence of Ino80 (Fig. 4E,F), we presume that the two phenomena are unrelated. The slight rescue of gap filling that we observed in htz1Δ swr1Δ double mutants after depletion of Ino80 in some experiments (Fig. EV4B,C) may have been due to the general suppression of the fitness defect of htz1Δ mutants by deletion of SWR1 (Morillo-Huesca et al, 2010). Overall, our data suggest that the role of the INO80 complex in DDT is distinct from its contribution to DSB repair.

## Chromatin dynamics contribute to multiple aspects of DNA replication stress tolerance

Our findings indicating that the INO80 complex acts downstream of PCNA ubiquitylation appear to contradict earlier reports about a contribution of the remodeller to Rad18 recruitment and PCNA ubiquitylation under conditions of high damage loads (Falbo et al, 2009). However, considering the notion that the bypass of tolerable levels of DNA damage proceeds predominantly in a postreplicative manner while high damage levels result in extended fork stalling and/or collapse, this apparent discrepancy may point to distinct actions of the INO80 complex at stalled or broken replication forks *versus* daughter-strand gaps.

The structure of nascent chromatin assembled immediately after passage of the replisome is expected to differ from that at later

stages of replication or in G2/M, perhaps calling for different remodelling activities for the purpose of lesion bypass. Consistent with this notion, recycling of parental histones H3 and H4 at the fork has been shown to impact pathway choice in replicative lesion processing (Dolce et al, 2022; González-Garrido and Prado, 2023). Intriguingly, it has also been reported that the INO80 complex prefers to slide hexasomes, lacking one H2A-H2B dimer, over conventional nucleosome core particles (Hsieh et al, 2022) and engages with hexasomes in a structurally distinct conformation (Wu et al, 2023; Zhang et al, 2023). A recent report proposed that the recycling of parental histones at the replisome, co-chaperoned by the FACT complex and Mcm2, involves histone hexamers (Li et al, 2024). This raises the possibility that the INO80 complex exhibits elevated or altered remodelling activity at nascent *versus* mature chromatin, based on the increased amount of extranucleosomal DNA associated with hexasomes.

Additional research is clearly needed for better insight into the role of the INO80 complex in replicative lesion processing. Specifically, it will be important to understand the crosstalk between the INO80 complex and other remodellers impinging on DDT, such as the BAF complex, which has been postulated to promote repriming in yeast and human cells (Niimi et al, 2012). Other factors have been implicated in DDT genetically, for example, the chromatin assembly factors CAF-1, for which an epistatic relationship was observed with the *RAD6* pathway (Game and Kaufman, 1999), and Asf1, which was reported to cooperate with Rad6 in PCNA ubiquitylation (Kats et al, 2009). Likewise, epigenetic modifications are important for DDT. For example, histone H4 acetylation appears to favour TLS over TS in yeast (Renaud-Young et al, 2015). In human cells, H4 acetylation defects compromise PCNA monoubiquitylation and Polymerase η recruitment in response to UV-induced DNA damage (Wong et al, 2011). Even certain histone mutants were reported to compromise the activity of TLS polymerases (Selvam et al, 2020), while others were shown to rescue the defects of DDT mutants in an INO80 complex-dependent manner (Hayashi et al, 2021). It is likely that many of these factors operate by modulating the accessibility of chromatin for DDT proteins. However, their precise mode of action as well as their temporal and spatial regulation remain to be elucidated.

# Methods

### Reagents and tools table

| Reagent/resource | Reference or source | Identifier or catalogue number |
|---|---|---|
| **Experimental models** | | |
| *S. cerevisiae* DF5 | (Finley et al, 1987) | N/A |
| *S. cerevisiae* W303 | (Thomas & Rothstein, 1989) | N/A |
| *S. cerevisiae* S288C | (Shimada et al, 2008) | N/A |
| DF5, W303, and S228C derivatives | This study | AppendixTable S1 |
| *Escherichia coli* DH10 MultiBac | (Sari et al, 2016) | N/A |

| Reagent/resource | Reference or source | Identifier or catalogue number |
|---|---|---|
| *Spodoptera frugiperda* Sf21 cells | Thermo Fisher Scientific | 11497013 |
| *Trichoplusia ni* High Five cells | Invitrogen | B85502 |
| **Recombinant DNA** | | |
| Plasmids | This study | Appendix Table S2 |
| **Antibodies** | | |
| Anti-FLAG, mouse monoclonal (M2) | Sigma-Aldrich Chemie | F1804 |
| Anti-FLAG, rabbit polyclonal | Sigma-Aldrich Chemie | F7425 |
| Anti-myc, mouse monoclonal (9E10) | N/A | In-house |
| Anti-Rad53, rabbit polyclonal | Abcam | ab104232 |
| Anti-PGK, mouse monoclonal (22C5D8) | Thermo Fisher Scientific | 459250 |
| Anti-α-Tubulin, rat monoclonal (YL1/2) | Sigma-Aldrich Chemie | 92092402 |
| Anti-ubiquitin, mouse monoclonal (P4D1) | Cell Signaling Technology | 3936 |
| Anti-Clb2, rabbit polyclonal | Santa Cruz Biotechnology | sc-9071 |
| Anti-PCNA, rabbit polyclonal | (Stelter and Ulrich, 2003) | N/A |
| Anti-Rad18, rabbit polyclonal | (Daigaku et al, 2010) | N/A |
| Anti-ssDNA, mouse monoclonal (16-19) | Millipore | MAB3034 |
| Anti-BrdU, mouse monoclonal (B44) | BD Biosciences | 347580 |
| Goat anti-mouse secondary antibody, Cy3-labelled | Abcam | AB6946 |
| Goat anti-mouse secondary antibody, HRP-labelled | Dako | P044701-2 |
| Donkey anti-mouse secondary antibody, IRDye800CW-labelled | LICOR Biosciences | 926-32230 |
| Donkey anti-mouse secondary antibody, IRDye680LT-labelled | LICOR Biosciences | 926-68022 |
| Goat anti-rabbit secondary antibody, HRP-labelled | Dako | P044801-2 |
| Donkey anti-rabbit secondary antibody, IRDye800CW-labelled | LICOR Biosciences | 926-32211 |
| Goat anti-rat secondary antibody, IRDye800CW-labelled | LICOR Biosciences | 926-32219 |
| **Oligonucleotides and other sequence-based reagents** | | |
| PCR primers | This study | Appendix Table S3 |
| **Chemicals, Enzymes and other reagents** | | |
| Alpha-factor (synthetic peptide: WHWLQLKPGQPM) | ProteoGenix | N/A |
| Nocodazole | APExBIO Technology | A8487 |
| Doxycycline hydrochloride | Sigma-Aldrich Chemie | D3447 |

| Reagent/resource | Reference or source | Identifier or catalogue number |
|---|---|---|
| Indole-3-acetic acid sodium salt (Auxin, IAA) | Biomol | Cay16954-10 |
| Methyl methanesulfonate (MMS) | Sigma-Aldrich Chemie | 129925 |
| 4-Nitroquinoline-1-oxide (4NQO) | Sigma-Aldrich Chemie | N8141 |
| Hydroxyurea (HU) | Sigma-Aldrich Chemie | H8627 |
| 5-Bromo-2-deoxyuridine (BrdU) | Sigma-Aldrich Chemie | B5002 |
| 5-ethynyl-2′-deoxyuridine (EdU) | Life Technologies | E10187 |
| Propidium iodide | Sigma-Aldrich Chemie | 81845 |
| 4′,6-Diamidine-2′-phenylindole dihydrochloride (DAPI) | Sigma-Aldrich Chemie | 10236276001 |
| YOYO-1 iodide | Thermo Fisher Scientific | Y-3601 |
| Sodium azide | Sigma-Aldrich Chemie | S2002 |
| RNase A | Sigma-Aldrich Chemie | 10109169001 |
| Proteinase K | Sigma-Aldrich Chemie | P4850 |
| Zymolyase 100 T | Carl Roth | 9329.2 |
| Concanavalin A | Sigma-Aldrich Chemie | L7647 |
| HF Polymerase | In-house | N/A |
| Lambda exonuclease | New England Biolabs | M0262L |
| NuPage LDS buffer | Invitrogen | NP0008 |
| Phusion HF buffer | New England Biolabs | B0518S |
| NuSieve low-melting point agarose | Lonza | 50081 |
| Ni-NTA beads | Qiagen | 30250 |
| HighPrep beads | Magbio | AC-60050 |
| ANTI-FLAG M2 Affinity Gel | Sigma-Aldrich Chemie | A2220_25mL |
| FLAG peptide | Biotrend | TP1274-25mg |
| Click-iT Plus imaging kit | Invitrogen | C10637 |
| MasterPure yeast DNA purification kit | Lucigen | MPY80200 |
| Hybond-N+ membranes | Cytiva | RPN203B |
| ECL detection kit | Cytiva | RPN2235 |
| Duolink in situ PLA probe anti-rabbit MINUS | Sigma-Aldrich Chemie | DUO92005 |
| Duolink in situ PLA probe anti-mouse PLUS | Sigma-Aldrich Chemie | DUO92001 |
| Duolink in situ detection reagents red | Sigma-Aldrich Chemie | DUO92008 |
| Duolink in situ mounting medium with DAPI | Sigma-Aldrich Chemie | DUO82040 |
| Human histone H2A | The Histone Source | HH2A |
| Human histone H2B | The Histone Source | HH2B |
| Human histone H3.3 | The Histone Source | HH3_3 |
| Human histone H4 | The Histone Source | HH4 |
| Lambda DNA | New England Biolabs | N3011SVIAL |
| Recombinant Albumin, Molecular Biology Grade | New England Biolabs | B9200SVIAL |
| **Software** | | |
| FlowJo v10.10 | BD Life Sciences | N/A |
| SoftWoRx v6.5.2 Release RC1 | GE Healthcare | N/A |
| LAS X version 3.7.6.25997 | Leica Microsystems | N/A |
| Image J FIJI 1.54p | https://fiji.sc/ (Schindelin et al, 2012) | N/A |
| Prism v7.0 | GraphPad | N/A |
| R v4.4.3 | https://www.R-project.org/ | N/A |
| Image Studio v3.1 | Licor | N/A |
| **Other** | | |
| Q125 sonicator | QSonica | Q125-110 |
| Novocyte Quanteon flow cytometer | Agilent | N/A |
| DeltaVision Elite widefield microscope | GE Healthcare | N/A |
| THUNDER imaging system | Leica Microsystems | N/A |
| Molecular combing system | Genomic Vision | ADMCS |
| Qubit fluorometer | Invitrogen | Q33226 |
| TYPHOON Imager | Cytiva | N/A |
| µ-Slide eight-well uncoated chambered glass bottom coverslips | Ibidi | 80821 |
| Engraved Combicoverslips | Genomic Vision | COV-002-RUO |
| HiTrap DEAE FF column, 1 ml | Cytiva | GE17-5055-01 |
| Resource Q column, 1 ml | Cytiva | GE17-1177-01 |
| Mono Q 5/50 GL column | Cytiva | GE17-5166-01 |
| HiLoad 16/600 Superdex 200 pg | Cytiva | 28989335 |
| Amicon Ultra concentrator, 30-kDa MWCO | Merck Millipore | UFC503008 |
| Amicon Ultra concentrator, 10-kDa MWCO | Merck Millipore | UFC801008 |
| NativePAGE Bis-Tris Mini Protein Gels, 3–12%, 1.0 mm | Invitrogen | BN1001BOX |

## Yeast strain construction and culture conditions

Cultures were grown in YPD at 30 °C unless otherwise noted. Strains carrying the *cdc13-1* allele were grown at 23 °C, treated with nocodazole for 2 h to induce a G2/M arrest before temperature shift to 37 °C to induce telomere deprotection (Fig. 3H). Strains carrying the *TET-RAD18* allele were cultured as described, using 2 µg/ml doxycycline for repression of the *TET* promoter where indicated

(Daigaku et al, 2010). Gene deletions and epitope- or degron-tagged alleles were generated with PCR-based methods (Janke et al, 2004; Morawska and Ulrich, 2013). To arrest cells in G1, α-factor was added at a concentration of 2 µg/ml for 1 h, and 4 µg/ml were added every subsequent hour. To release cells from G1 arrest, cells were washed twice with prewarmed water and resuspended in prewarmed YPD at 30 °C. MMS treatment was for 30 min with 0.02% for all cell cycle experiments (Figs. 2B–D, 3A–D,G, 4B,C,E, 5A,B and EV2B,E, EV3A–C,E, EV4A,D–H, EV5A) or 0.033% for Fig. 3F and all inducible damage bypass experiments (Figs. 3J–O, 4F and EV3F–H, EV4B,C). 4-nitroquinoline oxide was used at 0.1 µg/ml for 30 min in G1-arrested cells, and UV irradiation was with 20 J/m². For HU treatment, G1-arrested cells were released into YPD in the presence of 120 mM HU. To prevent entry into the next cell cycle, 15 µg/ml nocodazole from a stock of 1.5 mg/ml in DMSO was added to the culture 60 min after release (Figs. 2B–D, 3L–O, 4B,C,E,F, 5A,B and EV2B, EV2D,E, EV3F–H, EV4A–H, EV5A). For transient depletion of proteins carrying an auxin-inducible degron tag (AID*), 0.5 mM auxin was added at the start of G1 synchronisation (Figs. 2B–D, 3F,G, 4B,C,E and EV2D,E, EV3D,E, EV4A,D). Alternatively, 1 mM auxin was added during the final 45 min of G1 arrest before release (Figs. 3A–D and EV3A–C, EV4E–H). For all experiments involving gap filling in G2/M-arrested cells, Ino80 was depleted with 1 mM auxin for 1 h, and cultures were maintained in 0.5 mM auxin during the time course (Figs. 3K–O, 4F and EV3F–H, EV4B,C). BrdU was used at a concentration of 0.4 mg/ml. For spot assays, fivefold serial dilutions of exponential yeast cultures, starting with 0.14 OD$_{600}$, were spotted with a replica plater (0.125-inch diameter) onto YPD plates with or without DNA-damaging agents. To deplete proteins of interest for plate-based assays and control western blots, cultures were treated with 1 mM auxin for 1 h and/or with 2 µg/ml doxycycline for 6 h before spotting onto YPD plates containing 0.25 mM auxin and/or 2 µg/ml doxycycline (Fig. EV1A–D). Plates were incubated at 30 °C for 2 days or at relevant temperatures for the indicated time before imaging.

## Analysis of proteins in cell extracts and denaturing Ni-NTA pulldown

Total cell lysates were prepared by trichloroacetic acid (TCA) precipitation as described (Morawska and Ulrich, 2013), denatured in NuPage LDS buffer at 70 °C for 10 min with shaking at 1400 rpm, and resolved by SDS-PAGE followed by western blotting. Ubiquitylated forms of PCNA were isolated from strains carrying His-tagged PCNA by pulldown with Ni-NTA beads under denaturing conditions as described (Davies and Ulrich, 2012). Relevant proteins were detected by western blotting using antibodies listed in the Reagents and Tools Table.

## Cell cycle analysis

Cells were fixed in 70% ethanol at 4 °C for at least 1 h. Samples were washed twice with 50 mM sodium citrate, pH 7.0, incubated with 80 µg/ml RNase A at 50 °C for 1 h, followed by incubation with 80 µg/ml proteinase K at 50 °C for 1 h. Samples were sonicated with a Q125 sonicator equipped with an 8-tip horn for 10 cycles (2 s on/2 s off) at 70% output, and stained with 32 µg/ml propidium iodide.

DNA content was measured with a Novocyte Quanteon flow cytometer. Data were analysed with the FlowJo v10.10 software.

## Fluorescence microscopy

For imaging Rfa1$^{GFP}$, cells were fixed in 2.5% formaldehyde in 100 mM potassium phosphate, pH 6.4, for 10 min at room temperature, washed twice with 100 mM potassium phosphate, pH 6.6, and resuspended in 100 mM potassium phosphate, pH 7.4. Samples were plated on concanavalin A-coated chambered glass bottom coverslips, permeabilised with 70% ethanol for 30 min, washed once with 100 mM potassium phosphate, pH 7.4 and stained with 0.5 µg/ml DAPI for 10 min before imaging with a DeltaVision Elite system equipped with a 60× oil immersion objective (NA = 1.42), Scientific CMOS camera, InsightSSI solid state illumination and SoftWoRx software with built-in deconvolution algorithm. DAPI and GFP signals were obtained with DAPI and FITC filters, respectively.

## Analysis of ssDNA within native DNA fibres

ssDNA was visualised and quantified within native DNA fibres as described, with slight modifications (García-Rodríguez et al, 2018). Briefly, Ino80 was transiently depleted in the BrdU×7 background during G1 synchronisation, treated with MMS as indicated and released into S phase in the presence of 20 µM EdU. Cells (~10 OD$_{600}$) were inactivated by 0.1% sodium azide, spheroplasted with 0.1 mg/ml Zymolyase 100 T, embedded in 300 µl of 1% NuSieve low-melting point agarose and set for solidification in three plugs. Following proteinase K digestion at 37 °C for 2 days and extensive washing in 1× TE with 100 mM NaCl, genomic DNA was extracted from plugs by melting in 50 mM MES, pH 6.0, with 100 mM NaCl. Combing was performed on vinylsilane-coated coverslips with a molecular combing system. Single-stranded DNA was detected by immunostaining with mouse anti-ssDNA antibody and Cy3-labelled goat anti-mouse secondary antibody. EdU tracts were labelled with Alexa Fluor 647 via Click reaction. DNA fibres were counterstained with YOYO-1 and imaged with a THUNDER imaging system equipped with a FITC/TRITC/Cy5 filter set, 63× oil immersion objective (NA = 1.4), LED light source (LED8, Leica), a Leica DFC9000 GTC camera, and LAS X software.

Images were analysed with customised ImageJ macro scripts. Briefly, EdU and ssDNA signals were segmented by applying an auto-thresholding and an absolute thresholding, respectively. Identical parameters were chosen for each set of samples processed together, but parameters were adjusted individually for different sets of experiments. The lengths of individual EdU tracts, as well as the number of ssDNA tracts and the total length of ssDNA within each of these regions, were determined. The total percentage of ssDNA and the number of ssDNA tracts per unit length of DNA were then calculated.

## Daughter-strand gap filling assays in G2/M

To separate DDT from bulk genome replication, an inducible system was used to delay DDT to G2/M phase as described (Daigaku et al, 2010; García-Rodríguez and Ulrich, 2019). Briefly, strains carrying the *RAD18* allele under control of a tetracycline-repressible promoter (*TET-RAD18*) were grown overnight in YPD

in the presence of doxycycline to repress *RAD18* expression. Cells were then synchronised in G1 and treated with 0.033% MMS for 30 min before release into S phase in the presence of doxycycline but without MMS. Two hours after release, G2/M-arrested cells were treated with 1 mM auxin to degrade target proteins or left untreated for 1 h. Cells were then washed twice with YPD without doxycycline to induce Rad18 expression for gap filling. To prevent entry into the next cell cycle, cells were treated with 15 µg/ml nocodazole 1 h after Rad18 re-expression to maintain the G2/M arrest. As an additional precaution to prevent S phase entry of those cells that escaped the G2/M arrest, 10 µg/ml α-factor were additionally added to the culture every hour. To measure cell cycle recovery (Fig. 3K), nocodazole was omitted to allow progression to the next G1 phase. For analysing repair DNA synthesis, cells were incubated with 0.4 mg/ml BrdU 15 min before *RAD18* induction and resuspended with 0.4 mg/ml BrdU after washing out the doxycycline.

## Detection of BrdU incorporation

To quantify newly synthesised DNA in G2/M-arrested cells (Fig. 3N), genomic DNA was extracted from ca. 5 OD$_{600}$ cells for each time point using the MasterPure Yeast DNA purification kit according to the manufacturer's instructions. Genomic DNA (0.5 µg) in 3 µl TE was spotted onto Hybond-N+ membranes, air-dried and crosslinked with 1200 J/m$^2$ UV-C. Membranes were blocked in 5% skim milk in PBST, incubated with anti-BrdU antibody, followed by anti-mouse HRP antibody, and detected by ECL.

## Proximity ligation assays (PLA)

Approximately 1 OD$_{600}$ cells were fixed in 4% paraformaldehyde in 100 mM sodium phosphate, pH 7.6, for 15 min at room temperature, washed twice with 100 mM potassium phosphate, pH 7.4 and stored in the same buffer at 4 °C until further use. Samples were pre-incubated with 150 µl of Zymolyase buffer (1.2 M sorbitol, 100 mM potassium phosphate, pH 6.5) with 10 mM DTT for 10 min and then incubated with 150 µl of Zymolyase buffer with 10 mM DTT and 0.4 mg/ml Zymolyase 100 T for 30 min at 30 °C. Spheroplasting was monitored by the addition of 1% SDS to small aliquots and observing cell lysis under the microscope. Samples were washed twice with Zymolyase buffer, followed by centrifugation at 1000 × g for 5 min. Samples were then resuspended in 150 µl Zymolyase buffer (100 µl for G2-arrested cells to achieve a comparable cell density). Twelve-well Epoxy-coated slides were cleaned with a brush in distilled water, wiped with 70% ethanol and left to dry. Each well was coated with 0.1% poly-L-lysine for 10 min, washed with distilled water and left to dry completely. About 10 µl of spheroplast suspension were plated on each well, incubated for 1 h at room temperature, and gently washed three times in Zymolyase buffer with 5-min incubations. Cells were permeabilised by three incubations of 5 min each with 1% Triton in 100 mM potassium phosphate, pH 6.5. Samples were then blocked with blocking solution (Duolink in situ PLA probe kit) for 1 h at 37 °C and incubated with 10 µl primary antibody solution (10 µg/ml mouse anti-myc and 4 µg/ml rabbit anti-FLAG) overnight at 4 °C. PLA reactions were performed with Duolink in situ PLA probes and in situ detection reagents according to the manufacturer's

instructions, except that the ligation step was extended to 1 h at 37 °C. Slides were then mounted with Duolink in situ mounting medium with DAPI and imaged with a DeltaVision widefield microscope using DAPI and mCherry filters.

## Preparation of DNA substrates for nucleosome sliding assays

Control blunt-ended Widom 601 dsDNA with 80-bp extranucleosomal DNA in the 0N80 orientation (Kunert et al, 2022) was amplified by PCR, followed by purification using anion exchange chromatography on a 1 ml HiTrap DEAE FF column. Fractions containing DNA were dialysed to H$_2$O overnight, followed by concentration with a vacuum centrifugal concentrator.

To generate DNA substrates with single-stranded overhangs, specified DNA fragments were amplified by PCR according to the scheme in Fig. EV5B–E in 10 ml reactions with the indicated primer pairs, incorporating a 5′-phosphorylation on one and a 5′-fluorescein-labelled dT on the opposite end. Amplification was performed with HF DNA polymerase. DNA was precipitated with 0.3 M sodium acetate and 50% (v/v) isopropanol, resuspended in 0.5 ml H$_2$O, and purified by means of 1.8 volumes of HighPrep beads according to the manufacturer's protocol. DNA was eluted in 500 µl of 1× Lambda exonuclease buffer, quantified by Qubit and adjusted to a concentration of 300 ng/µl with 1× Lambda exonuclease buffer. A total of 0.02 U of Lambda exonuclease per µg of DNA was added to the reaction, followed by incubation at 37 °C for 3 h to digest the phosphorylated strand. Reactions were stopped by the addition of 60 mM EDTA. DNA was then precipitated on the HighPrep beads present in the reaction by addition of 0.6 volumes of a solution containing 50% polyethylene glycol (PEG 8000) and 1 M NaCl, washed twice with 80% ethanol, air-dried, resuspended in 0.5 ml H$_2$O and quantified by Qubit. Relevant ssDNA strands were annealed to produce the final DNA substrates with ssDNA overhangs.

## Nucleosome preparation

Human histones H2A, H2B, H3.3 and H4 were resuspended in an unfolding buffer (7 M guanidinium chloride, 20 mM Tris-HCl pH 7.5, 1 mM DTT) at a 1.1-fold excess of H2A and H2B and dialysed against four changes of 1 l volumes of refolding buffer (20 mM Tris pH 7.5, 2 M NaCl, 1 mM DTT, 0.5 mM EDTA pH 8.0) for 16 h at 4 °C. Histone octamers were purified by size exclusion chromatography using a Superdex 200 16/60 column. After concentrating to 4 mg/ml in centrifugal filters (Amicon Ultra, 10-kDa cutoff), histone octamers were stored in 50% glycerol at −20 °C. For reconstituting nucleosomes, control dsDNA or dsDNA containing a 38 bp single-stranded 5′- or 3′-overhang was mixed at a 1.1-fold molar excess with the histone octamer in a buffer containing 25 mM Tris-HCl, pH 7.5, 0.25 mM DTT and 2 M NaCl. The NaCl concentration was gradually decreased to 50 mM over 16 h at 4 °C by salt gradient dialysis. After this, nucleosomes were purified by anion exchange chromatography using a 1 ml Resource Q column, and fractions containing nucleosomes were pooled and dialysed to 25 mM Tris-HCl pH 7.5, 0.25 mM DTT and 50 mM NaCl, concentrated to 1 mg/ml (Amicon Ultra, 30-kDa cutoff), and stored at 4 °C or aliquoted and flash-frozen in liquid nitrogen in the presence of 10% of glycerol and stored at −80 °C. All buffers used

for nucleosomes assembled on substrates with ssDNA overhangs were supplemented with 1 mM EDTA to prevent degradation of ssDNA.

## Purification of *S. cerevisiae* INO80 complex from insect cells

Recombinant *S. cerevisiae* INO80 complex was purified as described previously (Oberbeckmann et al, 2021a), using two pFBDM vectors. One vector contained the coding sequences of the INO80 C-module subunits (C-terminally 2xFLAG tagged Ino80, Rvb1, Rvb2, Ies6 and Arp5) and a second vector contained the remaining coding sequences for the subunits of the A- and N-modules (Actin, Arp4, Arp8, Taf14, Ies2, Ies4, Ies1, Ies3, Ies5 and Nhp10). INO80ΔN, a complex where truncation of the Ino80 N-terminus (amino acids 1–461) causes a loss of the Nhp10 module, was produced analogously using one vector containing the coding sequences of C-terminally 2xFLAG tagged Ino80(Δ1–461), Rvb1, Rvb2, Ies6 and Arp5 and a second vector containing the remaining coding sequences for Actin, Arp4, Arp8, Taf14, Ies2 and Ies4 (Oberbeckmann et al, 2021b). Bacmids were generated using *E. coli* DH10 MultiBac cells. From each bacmid, baculoviruses were generated in *Spodoptera frugiperda* (Sf21) insect cells. Each baculovirus (1:200) was transferred to 1 l of *Trichoplusia ni* High Five culture for co-infection. Cells were cultured for 60 h at 27 °C and harvested by centrifugation at 4 °C.

For purification of the INO80 and INO80ΔN complexes, cells were resuspended in lysis buffer containing 50 mM Tris-HCl pH 7.9, 500 mM NaCl, 10% glycerol, 1 mM DTT, pepstatin A (0.28 µg/ml), PMSF (0.17 mg/ml) and benzamidine (0.33 mg/ml) and disrupted by sonication (3 × 1 min; duty cycle, 50%; and output control, 5). The lysate was cleared by centrifugation at 34,500 × g for 45 min at 4 °C. The supernatant was incubated with 3 ml of ANTI-FLAG M2 Affinity Gel for 1 h and washed with 50 ml of Wash 1 buffer (25 mM HEPES pH 8.0, 500 mM KCl, 10% glycerol, 0.05% IGEPAL CA630, 4 mM MgCl$_2$ and 0.25 mM DTT), 50 ml of Wash 2 buffer (25 mM HEPES pH 8.0, 200 mM KCl, 10% glycerol, 0.05% IGEPAL CA630, 4 mM MgCl$_2$ and 0.25 mM DTT), and 10 ml of buffer A (25 mM HEPES pH 8.0, 150 mM KCl, 2 mM MgCl$_2$ and 1 mM DTT). The protein was eluted from the matrix by incubation with 5 ml of buffer A supplemented with 0.2 mg/mL FLAG peptide in four incubation steps of 15 min each. The elution fractions were loaded onto a Mono Q 5/50 GL column and eluted by a linear salt gradient (150 mM KCl to 1 M KCl), resulting in a highly pure INO80 complex. Aliquots were flash-frozen in liquid nitrogen in the presence of 20% glycerol and stored at −80 °C.

## Nucleosome sliding assays

0N80 nucleosomes with 5′-fluorescein or 6'FAM–labelled extra-nucleosomal DNA with and without ssDNA overhangs were used for monitoring the sliding activity of purified recombinant INO80 complex. Nucleosome (150 nM) was incubated with 50 nM *S. cerevisiae* INO80 or INO80ΔN complex in sliding buffer (25 mM HEPES, pH 8, 60 mM KCl, 7% glycerol, 0.10 mg/ml recombinant albumin, 0.25 mM DTT] at 25 °C. The sliding reaction was started by the addition of 1 mM ATP/2 mM MgCl$_2$ and stopped at the indicated time points by the addition of Lambda DNA (0.2 mg/ml). Nucleosome species were separated by native polyacrylamide gel

electrophoresis (PAGE) on a 3 to 12% acrylamide Bis-Tris gel and visualised using the Typhoon imaging system. Experiments were performed in triplicate.

## Quantification and statistical analysis

Throughout this study, sample sizes were chosen based either on the standards of the field (RPA foci, DNA fibre analyses) or on the consistency of the results between biological replicates. Samples were blinded for DNA fibre analyses; otherwise, samples were not randomised or blinded. No samples were excluded from the analysis. *P* values are indicated in the graphs; for *p* < 0.0001, exact numbers are not provided by the analysis software (Prism v7.0). The number of biological replicates (*N*) is stated in the figure legends for each experiment. Note that even in cases where *N* = 1, all results were verified in the laboratory at least once in other settings or sample combinations.

RPA foci were quantified from microscopic images by analysis with ImageJ FIJI software (https://fiji.sc/) (Schindelin et al, 2012) using customised scripts (Wong et al, 2020) written in ImageJ macro language. All image analyses were performed on deconvolved images in 3D. Briefly, nuclear masks were created from DAPI signals by auto-thresholding. RPA foci were segmented by absolute thresholding. Numbers of RPA foci per nucleus were determined for a minimum of 100 cells per condition, and the total intensity of nuclear RPA foci was calculated by summing the intensities of all individual RPA foci for each nucleus. To compare statistical significance between two distributions at the end points of RPA foci kinetics during a cell cycle (Figs. 2D, 4C,E and EV4A,E,G), resolution after gap filling (Figs. 3M, 4F and EV3F,H, EV4B,C) and emergence (Figs. 3G and EV3E), the Mann–Whitney *U*-test was applied using GraphPad Prism v7.0.

Tracts of ssDNA in EdU-stained DNA fibres (Figs. 3C,D and EV3B,C) were analysed by determining the number and lengths of ssDNA tracts within the EdU tracts. A minimum of 114 EdU tracts were analysed per condition. The fraction of ssDNA within EdU-labelled DNA was calculated for each tract by dividing the total length of ssDNA within the EdU tract by the tract length. The ssDNA tract density was calculated by dividing the number of ssDNA tracts within individual EdU tracts by the total length of the respective tract. To compare statistical significance between two distributions, the Mann–Whitney test was applied using GraphPad Prism v7.0.

BrdU signal intensities (Fig. 3N) were quantified with Image Studio v3.1. All values were log$_{10}$- transformed, followed by background subtraction. To adjust for the varied signals between independent biological replicates performed on different days, each value was normalised to the overall mean value within a biological replicate (normalised BrdU signal). For statistical analysis, normality and homoscedasticity of the data were first confirmed by the Shapiro–Wilk normality and Bartlett tests, respectively. After confirming that experimental results were affected consistently for each replicate, we chose a linear model in R equivalent to paired *t*-tests for statistical analysis. We corrected for multiple testing using the multivariate t-distribution method implemented in the emmeans package (https://CRAN.R-project.org/package=emmeans).

Nucleosome sliding assays (Fig. 5D) were quantified in Image Studio by determining the intensities of the two bands corresponding to the different nucleosome positions after local background subtraction

(Fig. 5D) and plotting the intensity of the upper band (shifted nucleosome) as the percentage of the total intensity. One-phase decay was used for curve fitting in GraphPad Prism v7.0: $y(t) = (y_0 - p)e^{-Kt} + p$.

## Data availability

Customised scripts for the analysis of RPA foci (Wong et al, 2020) are available at GitHub (https://github.com/helle-ulrich-lab/image-analysis-PORTs) and Zenodo (https://doi.org/10.5281/zenodo.14930052).

The source data of this paper are collected in the following database record: biostudies:S-SCDT-10_1038-S44318-025-00580-4.

## Peer review information

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

## Acknowledgements

The authors thank Annika Brem, Susan Gasser and Katrin Paeschke for reagents and strains, IMB's Core Facilities for Flow Cytometry and Microscopy, as well as the Media Lab for their services and supplies, and Fridolin Kielisch from IMB's Bioinformatics Core Facility for detailed advice with statistical analyses. Funding for this work was provided by the Deutsche Forschungsgemeinschaft (DFG, German Research Foundation, project number 393547839 – SFB 1361).

## Author contributions

**Ronald P Wong**: Conceptualisation; Data curation; Formal analysis; Validation; Investigation; Methodology; Writing—original draft; Writing—review and editing. **Mariia Likhodeeva**: Formal analysis; Investigation; Writing—review and editing. **Karl-Peter Hopfner**: Supervision; Funding acquisition; Writing—review and editing. **Helle D Ulrich**: Conceptualisation; Data curation; Supervision; Funding acquisition; Writing—original draft; Project administration; Writing—review and editing.

Source data underlying figure panels in this paper may have individual authorship assigned. Where available, figure panel/source data authorship is listed in the following database record: biostudies:S-SCDT-10_1038-S44318-025-00580-4.

## Disclosure and competing interests statement

The authors declare no competing interests.

# Expanded View Figures

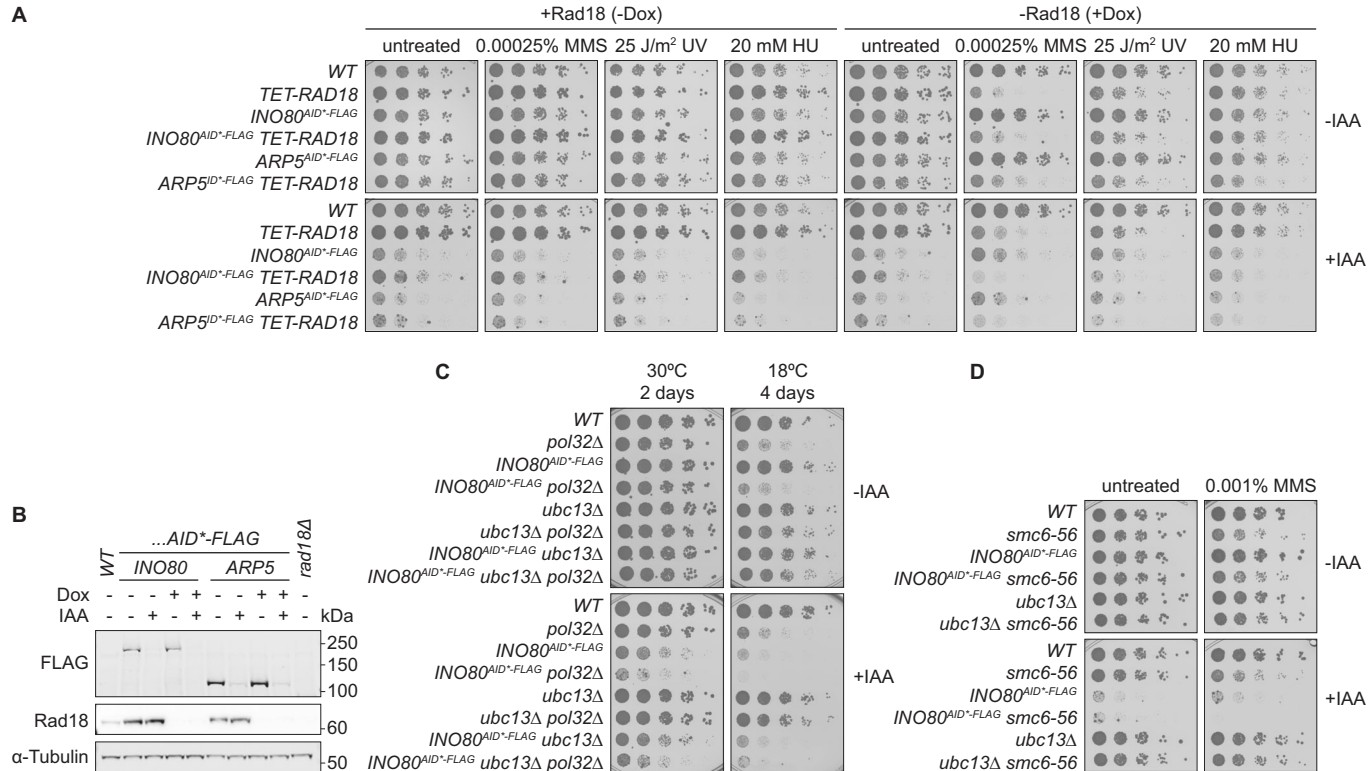

**Figure EV1. The INO80 complex contributes to *RAD6*-independent pathways of damage resistance.**

(A) Depletion of Ino80 or Arp5 via an auxin-inducible degron (AID*-FLAG) aggravates the DNA damage sensitivity induced by doxycycline-mediated repression of *RAD18* expression in the *TET-RAD18* strain background (Dox doxycycline, IAA auxin; biological replicates: $N = 1$). (B) Regulation of Ino80[AID*-FLAG], Arp5[AID*-FLAG] and Rad18 protein levels by auxin and doxycycline. Proteins were detected by western blotting in the strains used for the experiment shown in panel (A). A blot against α-tubulin served as a loading control (biological replicates: $N = 1$). (C, D) *INO80* does not act specifically in the TS pathway. Suppression of the cold sensitivity of *pol32Δ* was assessed upon auxin-mediated depletion of Ino80[AID*-FLAG] at the indicated temperatures (C; biological replicates: $N = 2$), and suppression of the damage sensitivity of *smc6-56* was assessed for mutant combinations by growth in the presence of MMS (D; biological replicates: $N = 1$). A *ubc13Δ* mutant served as a control for a TS-specific defect. Source data are available online for this figure.

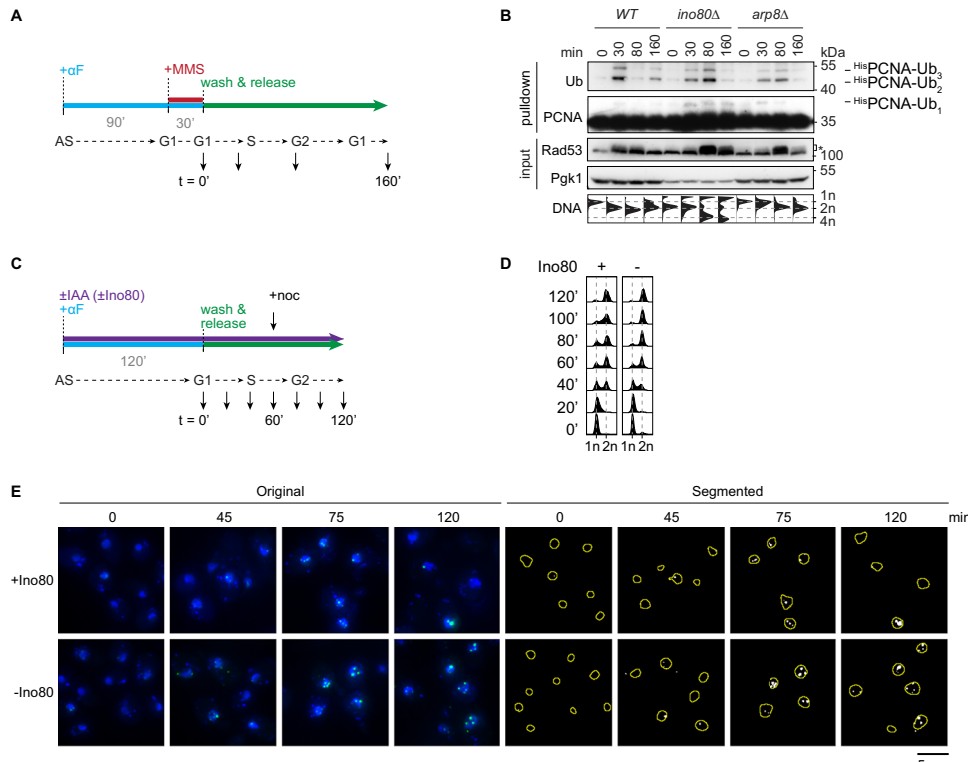

**Figure EV2. The INO80 complex prevents accumulation of ssDNA during replication of damaged DNA.**

(**A**) Experimental scheme to assess damage-induced PCNA ubiquitylation along S phase. Cells are synchronised in G1 phase and treated with 0.02% MMS for 30 min prior to release into S phase and collection of samples. (**B**) DNA damage-induced PCNA ubiquitylation is delayed but not reduced in the absence of functional INO80 complex. Modified PCNA was detected by Ni-NTA pulldown in lysates prepared under denaturing conditions from strains of the indicated genotypes harbouring a His$_6$-tagged allele of PCNA ($^{His}POL30$). Rad53 phosphorylation (*) was detected in total lysates as a marker of checkpoint activation, and Pgk1 served as a loading control. Corresponding cell cycle profiles are shown below the blots (biological replicates: $N = 1$). (**C**) Experimental scheme to assess cell cycle progression upon depletion of Ino80. Cells harbouring $INO80^{AID^*-FLAG}$ are synchronised in G1 in the presence or absence of auxin (IAA, 0.5 mM), washed and released into YPD medium. Nocodazole (noc) is added 60 min after release to prevent transition to the next cell cycle. Samples are collected for flow cytometry at the indicated time points. (**D**) Depletion of Ino80 causes minimal delay in cell cycle progression in the absence of DNA damage. Cell cycle profiles were determined by flow cytometry from samples treated as shown in panel (**C**) (biological replicates: $N = 1$). (**E**) An example of foci segmentation. Representative images of RPA foci from Fig. 2D, sampled during the course of an S phase in the presence or absence of Ino80, were segmented according to the procedure described in the Methods section (white: segmented foci; yellow: outline of nuclei). Source data are available online for this figure.

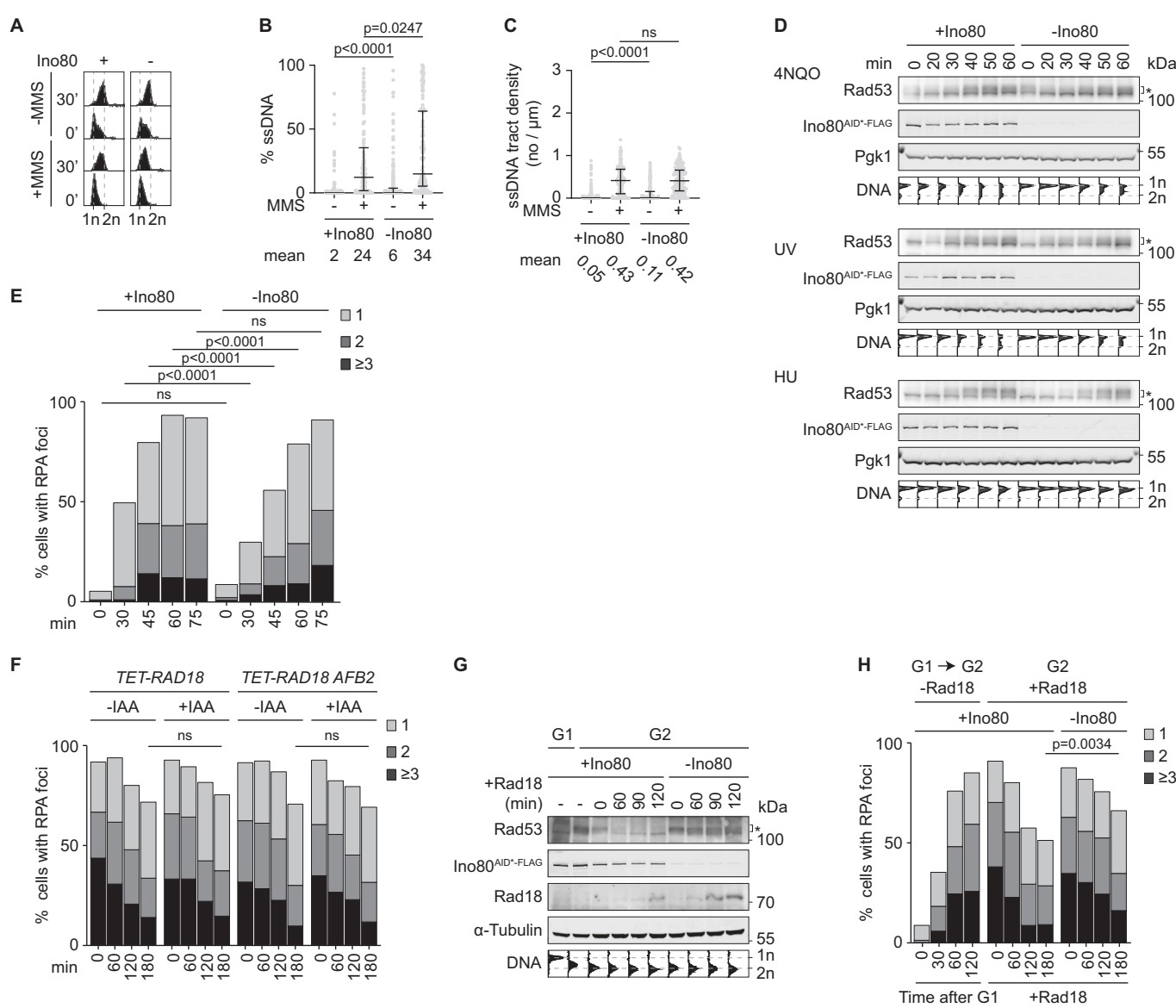

**Figure EV3. The INO80 complex facilitates both DNA resection and daughter-strand gap filling.**

(A–C) An independent biological replicate of the experiment shown in Fig. 3B–D (Mann–Whitney *U*-test; ns not significant; biological replicates: $N = 2$). Bars indicate median with interquartile ranges. (D) Depletion of Ino80 delays damage-induced checkpoint activation (*) but also S phase progression. Cells were treated as in Fig. 2A, but MMS was replaced with a 30 min treatment with 0.1 µg/ml 4-nitroquinoline oxide (4NQO) or a pulse of 20 J/m² UV-C before release or a release into medium containing 120 mM HU (biological replicates: $N = 2$). (E) Percentage of cells with RPA foci in the presence and absence of Ino80, corresponding to the experiment shown in Fig. 3G (Mann–Whitney *U*-test; ns not significant; biological replicates: $N = 2$). (F) Auxin and the auxin-responsive F-box protein Afb2 do not affect DDT-dependent resolution of RPA foci in G2/M. Gap-filling assays were performed as in Fig. 3M in the strains *TET-RAD18 RFA1^GFP* and *TET-RAD18 AFB2 RFA1^GFP* (Mann–Whitney *U*-test; ns not significant; biological replicates: $N = 1$). (G) An independent biological replicate of the experiment shown in Fig. 3L (biological replicates: $N = 2$). (H) An independent biological replicate of the experiment shown in Fig. 3M (Mann–Whitney *U*-test; biological replicates: $N = 2$). Source data are available online for this figure.

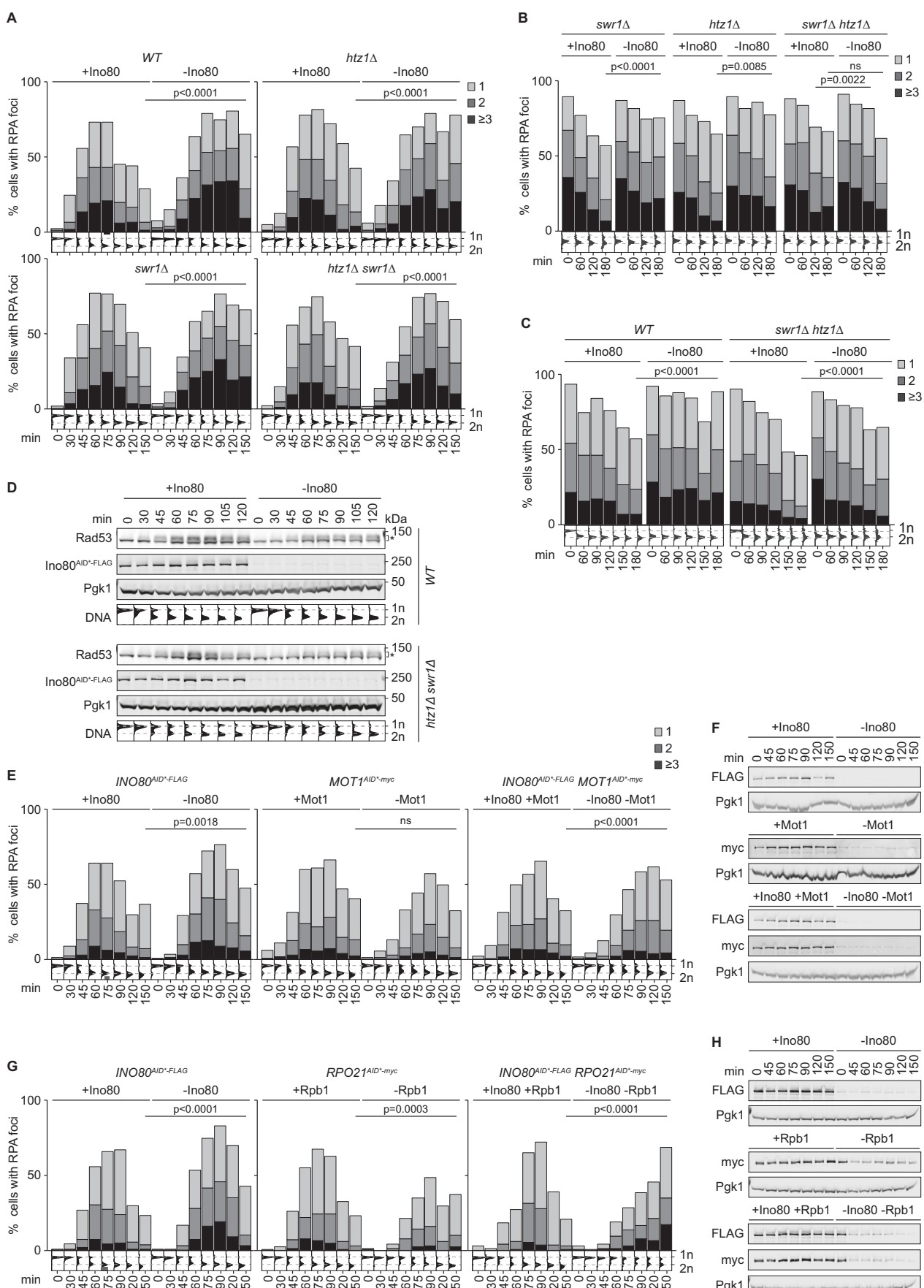

◀  **Figure EV4.  The function of the INO80 complex in postreplicative repair is unrelated to its role in H2A.Z extraction and repression of pervasive transcription.**

(A) An independent biological replicate of the experiment shown in Fig. 4E (Mann–Whitney *U*-test; biological replicates: $N = 2$). (B–C) Two independent biological replicates of the experiment shown in Fig. 4F (Mann–Whitney *U*-test; ns not significant; biological replicates: $N = 3$). (D) Deletion of *HTZ1* and *SWR1* does not suppress the prolonged checkpoint activation observed upon depletion of Ino80. *WT* and *htz1 swr1* mutants were treated as in Fig. 2A, and checkpoint activation was monitored via Rad53 phosphorylation (*). Cell cycle profiles are shown below the blots (biological replicates: $N = 1$). (E) Depletion of Mot1 causes cell cycle delays and interferes with the replication-associated emergence of RPA foci. RPA foci were monitored in the indicated strain backgrounds according to the scheme in Fig. 2A, except that AID*-tagged proteins were transiently degraded for 45 min before release into S phase (Mann–Whitney *U*-test; ns not significant; biological replicates: $N = 2$). Cell cycle profiles are shown below the graphs. (F) Western blots corresponding to the experiment shown in panel (E), demonstrating efficient depletion of Ino80 and Mot1 (biological replicates: $N = 2$). (G) Depletion of the large subunit of RNA polymerase II, Rpb1 (encoded by *RPO21*), delays replication but does not suppress the defect in the resolution of replication-associated RPA foci in the absence of Ino80. RPA foci were monitored in the indicated strain backgrounds according to the scheme in Fig. 2A, except that AID*-tagged proteins were transiently degraded for 45 min before release into S phase (Mann–Whitney *U*-test; biological replicates: $N = 1$). Cell cycle profiles are shown below the graphs. (H) Western blots corresponding to the experiment shown in panel (G), demonstrating efficient depletion of Ino80 and Rpb1 (biological replicates: $N = 1$). Source data are available online for this figure.

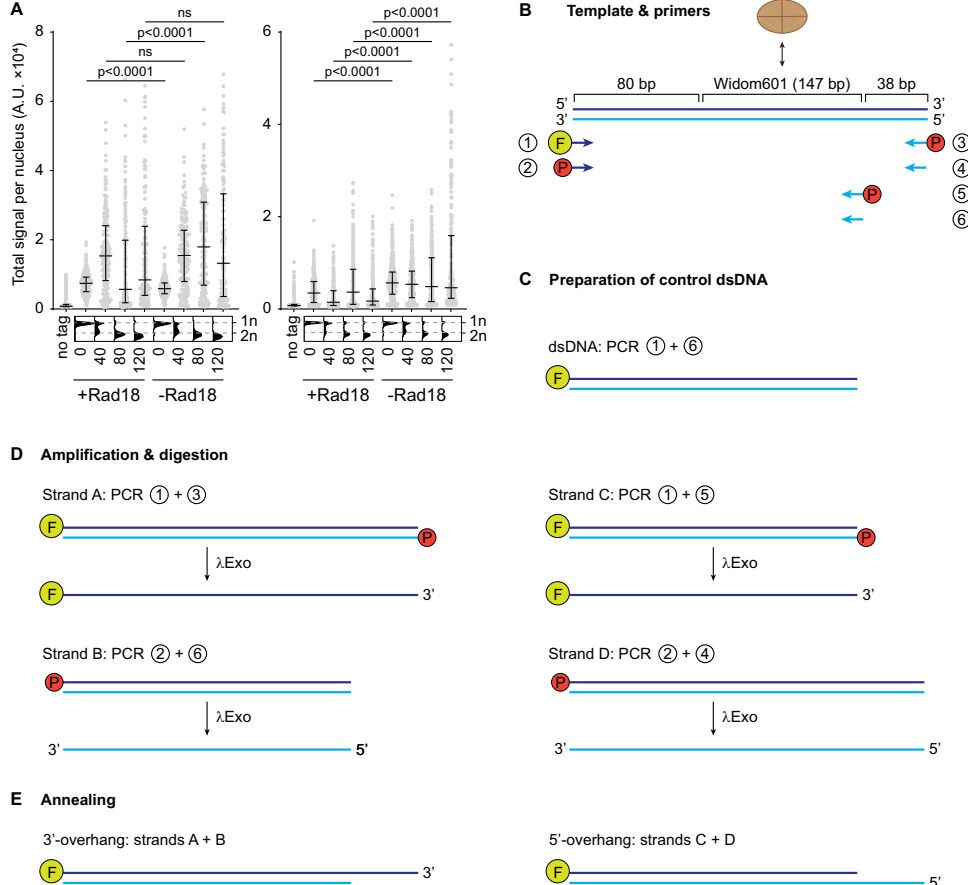

**Figure EV5. Construction of daughter-strand gap mimics for in vitro nucleosome sliding assays.**

(A) Two independent biological replicates of the proximity ligation assay shown in Fig. 5B (Mann–Whitney U-test; ns not significant; biological replicates: $N = 3$). Bars indicate median with interquartile ranges. (B) Template DNA for the preparation of substrates for nucleosome repositioning assays and oligonucleotides used for amplification of the template (P: 5′-phosphoryl; F: 5′-Fluoro-dT). (C) Preparation of control dsDNA substrate. (D, E) Preparation of substrates with 38 nucleotide overhangs requires isolation of individual strands (D) and annealing of single-stranded products in the desired combinations (E). PCR products produced with the indicated primer pairs were subjected to Lambda exonuclease (λExo) digestion to selectively degrade the 5′-phosphorylated strand. Source data are available online for this figure.

