## [Peer Review File · The EMBO Journal]

The INO80 chromatin remodeller facilitates DNA damage bypass via postreplicative gap repair

Ronald Wong, Mariia Likhodeeva, Karl-Peter Hopfner, and Helle Ulrich

Corresponding author(s): Helle Ulrich (h.ulrich@imb-mainz.de)

Review Timeline:

Submission Date:	23rd Apr 25
Editorial Decision:	26th May 25
Revision Received:	4th Sep 25
Accepted:	21st Sep 25

Editor: Hartmut Vodermaier

Transaction Report:

Prof. Helle D Ulrich
Institute of Molecular Biology (IMB)

-
Ackermannweg 4
Mainz 55128
Germany

26th May 2025

Re: EMBOJ-2025-121172
The INO80 chromatin remodeller facilitates DNA damage bypass via postreplicative gap repair

Dear Helle,

Thank you for submitting your study on INO80 roles in postreplicative gap repair for our consideration. Three expert referees have now evaluated it and returned the reports copied below. As you will see, all reviewers generally appreciate the importance and challenging nature of the question addressed here, as well as the interest of the presented findings. Pending adequate addressing of a number of well-taken technical concerns and more specific points, we would therefore be happy to consider a revised version further for EMBO Journal publication. Please note that referee 2 also raises a few follow-up issues regarding the direct molecular function of INO80 at ssDNA gaps, which would not be essential within the scope of this revision; however, including any discussions or data you may already have to expand on these questions would certainly be helpful.

Please be reminded, however, that our single-major-revision-round policy makes it important to diligently respond to each referee point at the time of resubmission; therefore, please do not hesitate to contact me ahead of resubmission with any questions you may have in this regard already. We would also be open to extension of the regular three-months revision period if needed; our 'scooping protection' (meaning that competing work appearing elsewhere in the meantime will not affect our considerations of your study) would of course remain valid also throughout such an extension.

Further information on preparing, formatting and uploading a revised manuscript can be found below and in our Guide to Authors. Thank you again for the opportunity to consider this work for The EMBO Journal, and I look forward to receiving your revised manuscript in due time.

With best regards,

Hartmut

9) To facilitate reproducibility and cross-laboratory adoption of methodologies, please structure the Materials & Methods section as outlined in our guide to authors, including a completed Reagents and Tools Table that can be downloaded from our author guidelines as well (<https://www.embopress.org/page/journal/14602075/authorguide#structuredmethods>).

10) Digital image enhancement is acceptable practice, as long as it accurately represents the original data and conforms to community standards. If a figure has been subjected to significant electronic manipulation, this must be clearly noted in the figure legend and/or the 'Materials and Methods' section. The editors reserve the right to request original versions of figures and the original images that were used to assemble the figure. Finally, we generally encourage uploading of numerical as well as gel/blot image source data; for details see: embopress.org/page/journal/14602075/authorguide#sourcedata

In the interest of ensuring the conceptual advance provided by the work, we recommend submitting a revision within 3 months (24th Aug 2025). Please discuss the revision progress ahead of this time with the editor if you require more time to complete the revisions. Use the link below to submit your revision:

Link Not Available

Referee #1:

The manuscript from Wong et al addresses the role of the INO80 chromatin remodeler in DNA damage bypass. This process can lead to the formation of a single stranded gap, which is repaired postreplicatively. In this study, the authors show that INO80 promotes postreplicative gap repair. They provide evidence that this is not dependent on H2A.Z exchange or related to modulating transcriptional responses to damage. They further suggest that INO80 promotes both gap expansion through Exo1-dependent resection as well as gap filling. Finally, they show that INO80 remodels a nucleosome away from single-strand/double-strand junctions in vitro, suggesting it can promote access to single stranded gaps.

This study addresses a complex question with challenging genetics, since these pathways are difficult to separate and INO80 has many different functions. Much of this work has been carefully performed and brings important understanding to these processes. In particular, the analyses supporting a contribution of INO80 to RAD6-independent as well as both branches of RAD6-dependent pathways and preventing accumulation of single stranded DNA in response to damage are very clear. It also provides important data suggesting that INO80 is working downstream of PCNA ubiquitylation in DNA damage tolerance. However, as outlined below, some conclusions are not yet supported well enough and require additional evidence.

The major issue in this manuscript is that conclusions rely heavily on analysis of cells with RPA foci as a measure of ssDNA

(particularly in Figures 3 and 4). There are two issues with this. First, the data as presented is insufficiently robust. Representative images of the stained cells with annotation of foci identification should be provided. In addition, the number of biological replicates performed should be stated in the legend for each assay. It states in the methods that Mann-Whitney tests were applied to compare statistical significance, but the test results do not appear to be annotated anywhere in the figures or figure legends (for example, Fig 3M). Instead, it seems a second biological replicate is provided in the supplementary figures. Second, an additional assay should be used to validate at least the key findings. This is important because in a few cases, the differences (or lack of differences) are not so clear (for example, the impact of *htz1* or *swr1* deletion on RPA foci profiles). Currently, the conclusions regarding the differences between INO80 roles in DNA damage tolerance and other pathways are not so strong.

Other issues and suggestions

1. In Figure 2, a FACS profile of + or - INO80 after release without MMS should be provided.
2. The figure panels were quite small, and when zooming in to look at the data, it appears there are some irregularities on the bar graphs (it looks like there are layers of graphs). This is probably an artifact of the PDF generation, but could the authors provide better images?
3. In Figure EV4, the authors use Rpb1 depletion to rule out removal of RNA pol II involvement in this process, but as they note, the data are a bit difficult to interpret due to the cell cycle impact. The authors could use transcription inhibitors instead to provide additional evidence for this conclusion.
4. The authors suggest that there is a separation of function between INO80 activity lacking Nhp10 compared with other subunits in genetic assays. Could they test a complex lacking Nhp10 in the in vitro remodelling assays to see whether this makes any difference to substrate specificity or activity?
5. In the gel shown in the remodelling assay (Fig 5B), the bands in the assay using a 5' overhang appear less discrete than the blunt and 3' overhang substrates. This is particularly true for the upper (remodelled) band and there seems to be a fair amount of signal in between the two species. Is this related to the preparation of this substrate or does it reflect a difference in positioning (perhaps more heterogeneity) when INO80 is using this substrate?

Referee #2:

General comment:

Using a combination of genetics, biochemistry and molecular biology in budding yeast, Wong et al. propose a function of the chromatin remodeller INO80 to facilitate gap repair through nucleosomes removal, thus facilitating access to the broken DNA matrix. They show that INO80 inactivation leads to negative genetic interactions when combined with mutants preventing the ubiquitination of PCNA in the presence of genotoxic agents, thus suggesting that INO80 plays extra functions in DNA damage tolerance (DDT) both in the template switching and TLS polymerases pathways. Furthermore, the authors show convincingly that INO80 inactivation does not prevent the accumulation of PCNA-ubi when cells face a moderate amount of DNA damage prior to S phase entry, a condition where DNA repair will mostly occur through post-replicative gap repair. This observation contrasts with the data presented in Falbo et al., 2009 where INO80 inactivation prevented PCNA-ubi. The authors clearly explain the differences in experimental setups, and propose that in the 2009 study, PCNA-ubi defects were confounded with a very strong DNA replication defect due to the high level of DNA damage maintained during the S phase. Then, Wong et al. demonstrate that INO80 favors gap repair by promoting (1) ssDNA generation through resection and (2) gap filling activity downstream of PCNA-ubi. Since INO80 is involved in multiple interactions that contribute to its numerous roles in cells, the authors made significant efforts to pinpoint/discard the relevant ones in the context of ssDNA gap repair. They could exclude: (1) a recruitment through γ H2A phosphorylation (as at DSB), (2) an involvement of H2A.Z, and tried their best to exclude the transcriptional silencing function of INO80 (3). Finally, they show that INO80 is able to move nucleosomes away from the ssDNA in vitro at 5' and 3'-overhang ssDNA-dsDNA junction, enabling them to propose a model where INO80 promote DNA repair at ssDNA gaps through its nucleosome sliding activity.

I found the study very well conducted. The methodology used is adapted and the data presented are well controlled and convincing. Introduction and discussions are well written and contain all the related references. In terms of impact, this study revises a model presented in a previous publication (Falbo et al., 2009) and propose a well-documented alternative to explain how INO80 promotes ssDNA gap repair. In my opinion the only weak point of the study is the lack of experimental evidences supporting the direct involvement of INO80 at ssDNA gaps in vivo. For instance, is INO80 recruited to sites of ssDNA gaps accumulation? This could be achieved by CHIP-qPCR but it requires to know where ssDNA gaps form and/or generate a well-defined DNA damage / ssDNA gap on the genome. It is also the case during the description of INO80's role in gap repair, the authors propose that INO80 facilitates gap repair by promoting resection and subsequent gap filling. I agree that the data presented in the study suggest that it is the case but they do not formally demonstrate it. For instance, does INO80 directly recruits Exo1 and/or promote its activity ? or is INO80 helping to recruit repair factors at ssDNA gaps (TS factors or TLS ?) ? Although, the suggested experiments will surely bring more mechanistic insights into INO80's function at ssDNA gaps, I fully acknowledge that they may be hard to achieve due to technical limitations and that they are not mandatory to sustain the model proposed by the authors.

I do not see any major issues with the proposed manuscript.

Minor points :

1. Missing information in figure legends (number of biological replicates, number of measures / replicates etc...). For Instance, Figure 3N does not display any information (type of error bars, number of replica, number of measures per replica etc...). It is also the case on other panels. Please add relevant information.
2. Is there any specific reason why the authors do not present a graph including the biological replicates altogether? It would help to visualize inter-experiments variation (for instance Fig 4, panels E and F and Fig EV4 A and B).
3. Some of the data presented by the authors show subtle changes in the control versus test condition (e.g. differences in the level of PCNA ubiquitination, differences in checkpoint inactivation kinetics,...). I know how sensitive these assays are and only the great quality of the blots presented allow to properly appreciate these variations. But, in order to help future readers, I would suggest to guide them by either adding some information in the text (e.g specific appearance of a non-phospho band for Rad53 in + INO80 condition) and/or directly on figures (e.g lines to show the non-phospho vs phospho form of Rad53 in Figure 3L, quantification of the Ubi levels overtime in Figure 3O?).
4. Results, page 4, first section. The description of genetic interactions is very general whereas some interactions are drug-specific. For instance, the authors indicate: "We found that *arp8Δ* sensitized both *ubc13Δ* and *tlsΔ* mutant...(Fig 1A)", a statement that is true for UV irradiation and MMS treatment but not visible under HU exposure. This does not alter the general conclusion, but I suggest to discuss the origin of these differences (ability of the drugs to induce DNA gaps ?)
5. Results, page 4, second section. The authors mention that the study from Falbo et al., 2009 used "heavy damage loads" to measure the impact of INO80 inactivation on PCNA ubi and want to check what is occurring under a moderate damage load. It would be informative for the reader to directly know in the present manuscript that the same dose of MMS (0.02%) was used in both studies but that the treatment was limited to 30 min in G1 for the present study, versus treatment with MMS all along S phase for Falbo et al., 2009.
6. Results, page 5, third section. The authors try to disentangle by which mechanism INO80 inactivation leads to the formation of longer ssDNA tracks and propose that it acts by promoting (1) DNA resection and (2) gap filling. I totally agree with the results presented but I'm not sure on the interpretation that INO80 mediates two different actions. If INO80 is required to promote DNA resection by *exo1* (by removing nucleosome), wouldn't it be expected to also show a defect in gap filling ? Did the authors check if an overexpression of *Exo1* rescues some of the INO80 mutants phenotype ? If an increase in *Exo1* rescues DNA resection but not gap filling, then it would be in favor of a dual effect of INO80 in the repair process.
7. Results, page 6, INO80 / H2A.Z ; the effect of *Arp5* inactivation is much stronger than *Arp8* depletion. Could the authors comment on that ?

Referee #3:

The yeast INO80 chromatin remodeling enzyme is known to play key roles in transcription, DNA double strand break repair, and the regulation of genome integrity checkpoint responses. Biochemical studies have defined two activities for INO80 - (1) it can use the energy of ATP hydrolysis to alter nucleosome positions in cis, using a ruler function to space nucleosomes, and (2) INO80 can evict the histone variant H2AZ from nucleosomes, replacing it with histone H2A. Previous work has indicated that this latter function of INO80 is employed during the repair of DNA double strand breaks, while the nucleosome spacing activity sets the position of the +1 nucleosome at gene promoters. Here the authors report that INO80 functions in post-replicative gap repair, promoting DNA damage bypass. Double mutant analysis indicates that INO80 works in a parallel pathway to the well-characterized RAD6 pathway for post-replicative repair. The authors make full use of both auxin-inducible degradation and doxycycline-mediated repression to control loss of function for gene products that influence cell cycle progression. ssDNA gap repair was followed by a powerful combination of DNA combing, PCNA-ubi, Rad53 phosphorylation, and RPA foci analyses. These data are convincing, and the authors' interpretation of the results are consistently balanced and thoughtful. Biochemical studies complemented the in vivo results, demonstrating that INO80 can reposition mononucleosomes from DNA end structures that mimic DNA gaps at forks.

In general, this is a solid manuscript with high quality data and appropriate methodologies. I only have a few minor comments:

1. In the introduction, it might be worth citing Adkins et al. (NSMB, 2013) who showed that *Exo1* resection is blocked by nucleosomes in vitro.
2. Page 5, middle. Here the authors discuss DNA combing results suggesting that ssDNA tracts are longer in the absence of INO80, but then they go on to show that RPA foci data indicates that there is less ssDNA in the absence of INO80. The end discussion concludes that INO80 stimulates resection. I assume these conflicting results are due to a combination of gap expansion and gap filling, but this needs to be reconciled.
3. Discussion, page 9. Here the authors indicate that INO80 may prefer to act on newly replicated chromatin, citing the preference of INO80 for hexosomes. Given that half of the nucleosomes behind the fork are modified with H3-K56Ac, it seems possible that this mark might also stimulate nucleosome sliding activities of INO80, not just H2AZ eviction (Watanabe et al., Science 2013).

Response to the Reviewers' Comments

We would like to thank the reviewers for their appreciation of our study and their constructive comments. In response to their points, we have now revisited the statistical analysis of our data and performed additional experiments to solidify our conclusions. Specifically, we show the segmentation of RPA foci, we have added an analysis of checkpoint activity via Rad53 phosphorylation to complement the RPA foci data, we performed proximity ligation assays (PLAs) to provide direct evidence for the physical presence of the INO80 complex at daughter-strand gaps, and we expanded our *in vitro* remodelling assays. Our specific responses to the reviewers' comments are detailed below.

Referee #1:

The manuscript from Wong et al addresses the role of the INO80 chromatin remodeller in DNA damage bypass. This process can lead to the formation of a single stranded gap, which is repaired postreplicatively. In this study, the authors show that INO80 promotes postreplicative gap repair. They provide evidence that this is not dependent on H2A.Z exchange or related to modulating transcriptional responses to damage. They further suggest that INO80 promotes both gap expansion through Exo1-dependent resection as well as gap filling. Finally, they show that INO80 remodels a nucleosome away from single-strand/double-strand junctions *in vitro*, suggesting it can promote access to single stranded gaps.

This study addresses a complex question with challenging genetics, since these pathways are difficult to separate and INO80 has many different functions. Much of this work has been carefully performed and brings important understanding to these processes. In particular, the analyses supporting a contribution of INO80 to RAD6-independent as well as both branches of RAD6-dependent pathways and preventing accumulation of single stranded DNA in response to damage are very clear. It also provides important data suggesting that INO80 is working downstream of PCNA ubiquitylation in DNA damage tolerance. However, as outlined below, some conclusions are not yet supported well enough and require additional evidence.

The major issue in this manuscript is that conclusions relay heavily on analysis of cells with RPA foci as a measure of ssDNA (particularly in Figures 3 and 4). There are two issues with this. First, the data as presented is insufficiently robust. Representative images of the stained cells with annotation of foci identification should be provided.

We now show a set of representative images and the result of the (published) segmentation procedure that is used for quantification (new Fig EV2E).

In addition, the number of biological replicates performed should be stated in the legend for each assay. It states in the methods that Mann-Whitney tests were applied to compare statistical significance, but the test results do not appear to be annotated anywhere in the figures or figure legends (for example, Fig 3M). Instead, it seems a second biological replicate is provided in the supplementary figures.

In response to the reviewer's concerns about the robustness of the data, we have thoroughly reassessed our statistical analysis throughout the manuscript and consulted a statistician about the validity of our approach. As a result, we have decided on a strategy that is adapted to the

type of data as outlined below. In addition, we indicate the type of test and the number of replicates in each figure legend.

- RPA foci profiles (Figs 2D, 4C, 4E, EV4A, EV4E, EV4G): These assays follow the number of RPA foci over the course of an S phase and generally show a biphasic behaviour of emergence and subsequent resolution of the foci. General cell cycle defects or replication problems cause a delay in the entire pattern, whereas gap expansion or filling defects cause a delay specifically in the emergence or resolution phase, respectively. In addition, slight variations in the experimental conditions during the release from G1 arrest can cause a shift in the overall temporal pattern, e.g. in the exact time at which RPA foci peak. Therefore, we find it more useful to only compare assays done together and provide flow cytometry data to correlate RPA foci kinetics with cell cycle progression. Where differences are subtle, for instance for the effect of *swr1Δ* *htz1Δ* mutant, we show multiple biological replicates. To strengthen our observation that depletion of Ino80 leads to accumulation of RPA foci, we now provide statistical significance values for the end point of the RPA foci profile using the Mann-Whitney test. Note that we show the data in categories (0, 1, 2, ≥ 3 foci per cell), but for statistical analysis we have taken into account the exact number of foci per cell. Significance level are now indicated on the respective graphs.
- RPA foci resolution (Figs 3M, EV3F, EV3H, 4F, EV4B-C): These assays differ from the above in that they only follow the resolution phase of the RPA foci. As above, we chose to compare the last time points of the assay to show the delay in RPA foci resolution upon Ino80 depletion, using the Mann-Whitney test. For the same reasons as above, we have performed these analyses only within a biological replicate and have not compared significance between replicates. We now indicate the significance level on the respective graphs.
- RPA foci emergence (Fig 3G, EV3E): This assay addresses specifically the emergence phase of the RPA signal. We chose to compare the individual time points over a relatively short time period between conditions performed together, using the Mann-Whitney test. Note that in these assays, inactivation of Ino80 merely delays RPA emergence, which is reflected by significant differences in RPA intensity during the intermediate time points (30, 45, 60 min), but not at the end of the time course (75 min).
- DNA fibre assays (Figs 3C-D, EV3B-C): In these assays, the percentage of ssDNA and the ssDNA tract density can be compared between strains or conditions in a straightforward manner using the Mann-Whitney test. Again, we are comparing data within a biological replicate and show a second biological replicate to provide additional robustness as we have collected samples at an arbitrary time point during S phase that might be subject to slight variation between experiments.
- BrdU incorporation assay (Fig 3N): This dot blot assay probes the newly synthesised DNA during gap filling. Based on the type of assay, only one value is generated per biological replicate. We observed that values from dot blot quantification vary greatly between different biological replicates, resulting in relatively large standard deviations, but each dataset obtained together was affected consistently. Therefore, based on the nature of the data, we chose a statistical model equivalent to paired t-test to compare the end point of the assay with or without Ino80.
- PLA assay (Figs 5B, EV5A): According to the standard in the field, samples were compared using the Mann-Whitney test within a biological replicate. Two additional

biological replicates are shown separately. In these assays, the overall variation between replicates is clearly detectable (thus making a combination of data problematic); yet the persistence of the PLA signal in the absence of Rad18 is consistent between replicates.

Second, an additional assay should be used to validate at least the key findings. This is important because in a few cases, the differences (or lack of differences) are not so clear (for example, the impact of *htz1* or *swr1* deletion on RPA foci profiles). Currently, the conclusions regarding the differences between INO80 roles in DNA damage tolerance and other pathways are not so strong.

To strengthen our conclusions about the relationship between the role of INO80 in DDT versus H2A.Z exchange by an independent assay, we have complemented the RPA foci profiles with time courses of Rad53 phosphorylation as an alternative way of monitoring gap filling via checkpoint deactivation (Fig EV4D). Together with new positive data showing a direct recruitment of the INO80 complex to sites of ssDNA (Figs 5A-B and EV5A, see also our response to reviewer #2), we now consider the overall support for our model rather strong.

Other issues and suggestions

1. In Figure 2, a FACS profile of + or - INO80 after release without MMS should be provided.

We now provide relevant FACS profiles (Fig EV2C-D).

2. The figure panels were quite small, and when zooming in to look at the data, it appears there are some irregularities on the bar graphs (it looks like there are layers of graphs). This is probably an artifact of the PDF generation, but could the authors provide better images?

We did not detect any flaws in the bar graphs, but we are providing the images as separate files and hope that the reviewers have access to high-resolution versions if needed.

3. In Figure EV4, the authors use Rpb1 depletion to rule out removal of RNA pol II involvement in this process, but as they note, the data are a bit difficult to interpret due to the cell cycle impact. The authors could use transcription inhibitors instead to provide additional evidence for this conclusion.

The use of transcription inhibitors in budding yeast is not straightforward. Transcription inhibitors commonly used in mammalian cells, such as α -amanitin and DRB, are not effective due to poor uptake and lack of conservation of the relevant targets in budding yeast. Transcription inhibitors commonly used in budding yeast, such as thiolutin and 1,10-phenanthroline, were recently found to have pleiotropic effects *in vivo* which would complicate our analysis [Qiu *et al*, *Nucleic Acids Res* 52: 2546 (2024); McNeil *et al*, *EMBO Rep* 25: 68 (2024)]. Therefore, we decided against using transcription inhibitors in this study.

4. The authors suggest that there is a separation of function between INO80 activity lacking Nhp10 compared with other subunits in genetic assays. Could they test a complex lacking Nhp10 in the *in vitro* remodelling assays to see whether this makes any difference to substrate specificity or activity?

We have now tested the activity of an INO80 complex lacking the Nhp10 module in the sliding assays (Fig 5D-E). We found no defect on any of the substrates; in fact, sliding activity tended to be enhanced. This phenomenon has been observed before on blunt-ended substrates by the

Narlikar lab [Zhou *et al*, *Mol Cell* 69: 677 (2018)]. Thus, the Nhp10 module appears to contribute neither *in vivo* nor *in vitro* to gap processing, in support of our model.

5. In the gel shown in the remodelling assay (Fig 5B), the bands in the assay using a 5' overhang appear less discrete than the blunt and 3' overhang substrates. This is particularly true for the upper (remodelled) band and there seems to be a fair amount of signal in between the two species. Is this related to the preparation of this substrate or does it reflect a difference in positioning (perhaps more heterogeneity) when INO80 is using this substrate?

We thank the reviewer for pointing this out. We cannot fully exclude a degree of heterogeneity in the INO80 activity. In addition, however, repositioning activity is much more difficult to detect on substrates carrying ssDNA overhangs compared to blunt-ended substrates. The 5'-overhang seems to be particularly problematic. Due to the single-stranded overhang, the difference in running behaviour between starting and remodelled substrate is smaller, and the propensity of the single-stranded region to adopt secondary structures could cause suboptimal running behaviour in the native polyacrylamide gels used for the assays. Moreover, we observed some batch-to-batch variation in the quality of the nucleosome loading, particularly with the overhang-containing substrates. In the course of our experiments using the INO80 Δ N mutant, we also repeated the original assays, and we now show the quantifications on the different substrates in separate graphs (see also our response to reviewer #3, point 3).

Referee #2:

General comment:

Using a combination of genetics, biochemistry and molecular biology in budding yeast, Wong et al. propose a function of the chromatin remodeller INO80 to facilitate gap repair through nucleosomes removal, thus facilitating access to the broken DNA matrix. They show that INO80 inactivation leads to negative genetic interactions when combined with mutants preventing the ubiquitination of PCNA in the presence of genotoxic agents, thus suggesting that INO80 plays extra functions in DNA damage tolerance (DDT) both in the template switching and TLS polymerases pathways. Furthermore, the authors show convincingly that INO80 inactivation does not prevent the accumulation of PCNA-ubi when cells face a moderate amount of DNA damage prior to S phase entry, a condition where DNA repair will mostly occur through post-replicative gap repair. This observation contrasts with the data presented in Falbo et al., 2009 where INO80 inactivation prevented PCNA-ubi. The authors clearly explain the differences in experimental setups, and propose that in the 2009 study, PCNA-ubi defects were confounded with a very strong DNA replication defect due to the high level of DNA damage maintained during the S phase. Then, Wong et al. demonstrate that INO80 favors gap repair by promoting (1) ssDNA generation through resection and (2) gap filling activity downstream of PCNA-ubi. Since INO80 is involved in multiple interactions that contribute to its numerous roles in cells, the authors made significant efforts to pinpoint/discard the relevant ones in the context of ssDNA gap repair. They could exclude: (1) a recruitment through γ H2A phosphorylation (as at DSB), (2) an involvement of H2A.Z, and tried their best to exclude the transcriptional silencing function of INO80 (3). Finally, they show that INO80 is able to move nucleosomes away from the ssDNA *in vitro* at 5' and 3'-overhang ssDNA-dsDNA junction, enabling them to propose a model where INO80 promote DNA repair at ssDNA gaps through its nucleosome sliding activity.

I found the study very well conducted. The methodology used is adapted and the data presented are well controlled and convincing. Introduction and discussions are well written and contain all the related references. In terms of impact, this study revises a model presented in a previous publication (Falbo et al., 2009) and propose a well-documented alternative to explain how INO80 promotes ssDNA gap repair. In my opinion the only weak point of the study is the lack of experimental evidences supporting the direct involvement of INO80 at ssDNA gaps *in vivo*. For instance, is INO80 recruited to sites of ssDNA gaps accumulation? This could be achieved by ChIP-qPCR but it requires to know where ssDNA gaps form and/or generate a well-defined DNA damage / ssDNA gap on the genome.

This is an important point. To follow up on the reviewer's suggestion, we have performed proximity ligation assays (PLAs) to monitor the physical presence of the INO80 complex at sites of ssDNA formed in response to DNA damage. Unlike ChIP-qPCR, PLAs do not require knowledge about the location of the interaction. As shown in Figs 5A-B and EV5A, we indeed observe a co-localisation of Ino80^{FLAG} with Rfa1^{9myc} that is prolonged in the absence of Rad18, i.e. when gaps accumulate. Given that under the experimental conditions, RPA foci represent postreplicative daughter-strand gaps, we consider this strong evidence for a direct physical involvement of INO80 at the gaps.

It is also the case during the description of INO80's role in gap repair, the authors propose that INO80 facilitates gap repair by promoting resection and subsequent gap filling. I agree that the data presented in the study suggest that it is the case but they do not formally demonstrate it. For instance, does INO80 directly recruits Exo1 and/or promote its activity ? or is INO80 helping to recruit repair factors at ssDNA gaps (TS factors or TLS ?) ? Although, the suggested experiments will surely bring more mechanistic insights into INO80's function at ssDNA gaps, I fully acknowledge that they may be hard to achieve due to technical limitations and that they are not mandatory to sustain the model proposed by the authors.

We appreciate the reviewer's suggestions regarding possible mechanisms by which the INO80 complex might facilitate gap expansion. We feel that the assays addressing these scenarios go beyond the scope of the current manuscript; however, as suggested by reviewer #3, we now cite Adkins *et al* [*Nat Struct Mol Biol* 20: 836 (2013)], who reported that Exo1-mediated resection is blocked by nucleosomes *in vitro*. A mechanism where the INO80 complex slides nucleosomes out of the way to facilitate Exo1 activity is therefore plausible and consistent with our data, as we outline in the discussion.

I do not see any major issues with the proposed manuscript.

Minor points :

1. Missing information in figure legends (number of biological replicates, number of measures / replicates etc...). For Instance, Figure 3N does not display any information (type of error bars, number of replica, number of measures per replica etc...). It is also the case on other panels. Please add relevant information.

We apologise for the omission and have now provided information on all quantifications shown and the number of replicates that were performed and/or shown. Please see also our response to reviewer #1 for a detailed description of our statistical analysis.

2. Is there any specific reason why the authors do not present a graph including the biological replicates altogether? It would help to visualize inter-experiments variation (for instance Fig 4, panels E and F and Fig EV4 A and B).

As explained in response to reviewer #1, we decided against combining biological replicates of RPA foci kinetics but show them separately in most of the experiments. As explained above, we cannot guarantee that the exact timing of the sampling is identical across all replicates. As small differences would blur the overall time course of RPA foci if replicates were averaged, we prefer not to combine data from independent replicates. Importantly, all replicates show qualitatively the same effect, and we have now added statistical significance values to the end points of each of these assays.

3. Some of the data presented by the authors show subtle changes in the control versus test condition (e.g. differences in the level of PCNA ubiquitination, differences in checkpoint inactivation kinetics,...). I know how sensitive these assays are and only the great quality of the blots presented allow to properly appreciate these variations. But, in order to help future readers, I would suggest to guide them by either adding some information in the text (e.g. specific appearance of a non-phospho band for Rad53 in + INO80 condition) and/or directly on figures (e.g. lines to show the non-phospho vs phospho form of Rad53 in Figure 3L, quantification of the Ubi levels overtime in Figure 3O?).

To facilitate the interpretation of the Rad53 gels, we have now marked the phosphorylated forms of Rad53 by an asterisk. We have also added explanations to the main text at the first occurrence of this type of blot and in the legend to Fig 3L, where the non-phosphorylated form of Rad53 is depleted due to the strong checkpoint activation and the pattern therefore looks different. Ubiquitylated forms of PCNA are indicated in the figures, but their quantification from biological replicates is problematic due to the same reasons that we mention above for RPA foci profiles.

4. Results, page 4, first section. The description of genetic interactions is very general whereas some interactions are drug-specific. For instance, the authors indicate: "We found that *arp8Δ* sensitized both *ubc13Δ* and *tlsΔ* mutant...(Fig 1A)", a statement that is true for UV irradiation and MMS treatment but not visible under HU exposure. This does not alter the general conclusion, but I suggest to discuss the origin of these differences (ability of the drugs to induce DNA gaps ?)

This is a valid point, and we thank the reviewer for the suggestion to expand on it. We envision that the synergy between TS and TLS defects and INO80 complex mutants with respect to MMS and UV, but not HU indeed stems from the site at which replication stress is sensed and processed: we previously showed that HU causes checkpoint activation at stalled forks, whereas MMS and UV activate damage signalling in daughter-strand gaps in a process that requires Exo1 activity (García-Rodríguez *et al*, *EMBO J* 2018). The partial defects of TS and TLS mutants to repair daughter-strand gaps may therefore aggravate the defect imparted by INO80 deficiency. We now comment on this effect in the Results and the first paragraph of the Discussion section.

5. Results, page 4, second section. The authors mention that the study from Falbo *et al.*, 2009 used "heavy damage loads" to measure the impact of INO80 inactivation on PCNA ubi and want to check what is occurring under a moderate damage load. It would be informative for the reader to directly know in the present manuscript that the same dose of MMS (0.02%) was used

in both studies but that the treatment was limited to 30 min in G1 for the present study, versus treatment with MMS all along S phase for Falbo et al., 2009.

Thank you for this suggestion; we now explicitly compare the conditions between the two studies.

6. Results, page 5, third section. The authors try to disentangle by which mechanism INO80 inactivation leads to the formation of longer ssDNA tracks and propose that it acts by promoting (1) DNA resection and (2) gap filling. I totally agree with the results presented but I'm not sure on the interpretation that INO80 mediates two different actions. If INO80 is required to promote DNA resection by *exo1* (by removing nucleosome), wouldn't it be expected to also show a defect in gap filling ? Did the authors check if an overexpression of *Exo1* rescues some of the INO80 mutants phenotype ? If an increase in *Exo1* rescues DNA resection but not gap filling, then it would be in favor of a dual effect of INO80 in the repair process.

The reviewer is correct that the results obtained for INO80 contributions to gap filling and gap expansion appear contradictory; however, they both eventually promote gap repair. We propose that at the "left" end of the gap, INO80 facilitates gap filling (by TLS), while at the "right" end it facilitates gap expansion (by *Exo1*). The net effect would still be repair (= filling) of the gaps because gap expansion is a prerequisite for the TS pathway, which subsequently fills gaps by homologous recombination. We have tried to make this clearer in our discussion of the results. We are sure that *Exo1* overexpression would affect the kinetics of these processes and probably their relative balance; however, it is difficult to predict whether it would result in a net slow-down of gap repair (due to excessive expansion) or a net acceleration (due to more efficient TS).

7. Results, page 6, INO80 / H2A.Z ; the effect of *Arp5* inactivation is much stronger than *Arp8* depletion. Could the authors comment on that ?

These results are consistent with recently published observations that *Arp5* is more important for core sliding activity, while *Arp8* has a regulatory role in monitoring the length of flanking DNA and even exerts an autoinhibitory effect on the sliding *in vitro* [Kunert *et al*, *Sci Adv* 8: eadd3189 (2022); Kaur *et al*, *Science* 389: eadr3831 (2025)].

Referee #3:

The yeast INO80 chromatin remodeling enzyme is known to play key roles in transcription, DNA double strand break repair, and the regulation of genome integrity checkpoint responses. Biochemical studies have defined two activities for INO80 - (1) it can use the energy of ATP hydrolysis to alter nucleosome positions in cis, using a ruler function to space nucleosomes, and (2) INO80 can evict the histone variant H2AZ from nucleosomes, replacing it with histone H2A. Previous work has indicated that this latter function of INO80 is employed during the repair of DNA double strand breaks, while the nucleosome spacing activity sets the position of the +1 nucleosome at gene promoters. Here the authors report that INO80 functions in post-replicative gap repair, promoting DNA damage bypass. Double mutant analysis indicates that INO80 works in a parallel pathway to the well-characterized RAD6 pathway for post-replicative repair. The authors make full use of both auxin-inducible degradation and doxycycline-mediated repression to control loss of function for gene products that influence cell cycle progression. ssDNA gap repair was followed by a powerful combination of DNA combing, PCNA-ub, Rad53

phosphorylation, and RPA foci analyses. These data are convincing, and the authors' interpretation of the results are consistently balanced and thoughtful. Biochemical studies complemented the in vivo results, demonstrating that INO80 can reposition mononucleosomes from DNA end structures that mimic DNA gaps at forks.

In general, this is a solid manuscript with high quality data and appropriate methodologies. I only have a few minor comments:

1. In the introduction, it might be worth citing Adkins et al. (NSMB, 2013) who showed that Exo1 resection is blocked by nucleosomes in vitro.

We now cite this paper in the discussion when integrating our model with previously published data.

2. Page 5, middle. Here the authors discuss DNA combing results suggesting that ssDNA tracts are longer in the absence of INO80, but then they go on to show that RPA foci data indicates that there is less ssDNA in the absence of INO80. The end discussion concludes that INO80 stimulates resection. I assume these conflicting results are due to a combination of gap expansion and gap filling, but this needs to be reconciled.

As explained in our response to reviewer #2 (point 6), the two opposing effects are only seemingly contradictory, as eventually they both promote gap repair (one via TLS, the other via TS). We have now expanded on this point in the first paragraph of the Discussion section.

3. Discussion, page 9. Here the authors indicate that INO80 may prefer to act on newly replicated chromatin, citing the preference of INO80 for hexosomes. Given that half of the nucleosomes behind the fork are modified with H3-K56Ac, it seems possible that this mark might also stimulate nucleosome sliding activities of INO80, not just H2AZ eviction (Watanabe et al., Science 2013).

To address this interesting point, we loaded nucleosomes containing a histone H3 acetylation-mimicking mutant, K56Q, onto a blunt-ended substrate and performed sliding assays. We found that the INO80 complex is capable of sliding this mutant nucleosome (see figure below), and we do not see a significant difference to canonical nucleosomes. However, it should be noted that quantifying relative activities on different substrates in the sliding assay is difficult because the sliding activity varies with the quality of the nucleosome preparation, in particular by the amount of residual free DNA that is inhibitory. Thus, assays performed with different nucleosome preparations (i.e., in this case H3-wt versus H3-K56Q) are therefore difficult to properly quantify and not very meaningful for small differences. For this reason, we did not extend these assays to the substrates containing single-stranded overhangs (which are even more difficult to prepare) and would prefer not to show the data on the blunt-ended substrate in the manuscript.

Figure for the reviewer. Mutant nucleosomes mimicking H3-K56 acetylation (H3^{K56Q}) are mobilised by the INO80 complex. Sliding assays were performed as described in the manuscript and shown in Figure 5. **A** Representative images of the gels used for analysis. **B** Quantification from three independent experiments.

Prof. Helle D Ulrich
Institute of Molecular Biology (IMB)

-
Ackermannweg 4
Mainz 55128
Germany

21st Sep 2025

Re: EMBOJ-2025-121172R
The INO80 chromatin remodeller facilitates DNA damage bypass via postreplicative gap repair

Dear Helle,

Thank you for submitting your final revised manuscript for our consideration. I am pleased to inform you that following positive re-review by one of the original referees (see below), we have now accepted it for publication in The EMBO Journal.

With kind regards,

Hartmut

Referee #1:

The revised manuscript from Wong et al does an excellent job addressing the issues raised. The major issue was whether the RPA foci dynamics were sufficiently robust to support the conclusions. The differences observed here are in some cases subtle and the timing is not consistent between replicates making conclusions difficult, but the authors have addressed this thoroughly with the inclusion of robust statistical analysis and additional data. This is an excellent study that sheds light on the role of INO80 in DNA damage bypass.